# Brain meta-state transitions demarcate thoughts across task contexts exposing the mental noise of trait neuroticism

Julie Tseng [1,2,5] & Jordan Poppenk [1,3,4,5 ✉]

Researchers have observed large-scale neural meta-state transitions that align to narrative events during movie-viewing. However, group or training-derived priors have been needed to detect them. Here, we introduce methods to sample transitions without any priors. Transitions detected by our methods predict narrative events, are similar across task and rest, and are correlated with activation of regions associated with spontaneous thought. Based on the centrality of semantics to thought, we argue these transitions serve as general, implicit neurobiological markers of new thoughts, and that their frequency, which is stable across contexts, approximates participants' mentation rate. By enabling observation of idiosyncratic transitions, our approach supports many applications, including phenomenological access to the black box of resting cognition. To illustrate the utility of this access, we regress resting fMRI transition rate and movie-viewing transition conformity against trait neuroticism, thereby providing a first neural confirmation of mental noise theory.

[1] Centre for Neuroscience Studies, Queen's University, Kingston, ON K7L 3N6, Canada. [2] Neurosciences and Mental Health, The Hospital for Sick Children, Toronto, ON M5G 1X8, Canada. [3] Department of Psychology, Queen's University, Kingston, ON K7L 3L3, Canada. [4] School of Computing, Queen's University, Kingston, ON K7L 2N8, Canada. [5] These authors contributed equally: Julie Tseng, Jordan Poppenk. ✉email: jpoppenk@queensu.ca

For a long time, the only window to the mind was through introspection, which posed a methodological problem due to the unreliable and disruptive nature of meta-cognition (i.e., thinking about one's own thoughts[1,2]). However, technological advances in brain imaging have allowed researchers to uncover the contents of thought directly from neural signals; researchers today can readily decode object categories (e.g., faces and houses) from spatial patterns in participants' functional magnetic resonance imaging (fMRI) data[3], and have even used spatial patterns in visual areas during sleep to reconstruct dream imagery[4].

In addition to what people are thinking about, researchers are also increasingly interested in how we think. For example: how does consciousness flow continuously from one thought to the next (i.e., the flights and perches of thought[5,6] or spontaneous thought[7]). In spontaneous thought research, a thought is defined as a mental state or sequence of mental states, and a mental state is defined as a "transient cognitive or emotional state of the organism that can be described in terms of its contents and the relation that the subject bears to the contents (for example, perceiving, believing, fearing, imagining, or remembering)"[7].

As most definitions of a thought concern its contents, we propose that implicit measurement of changes in semantic content offers an interesting alternative to meta-cognition. Accordingly, one could infer a new thought has arisen in a participant by observing a switch from one active category to another. One important reason this strategy has not been deployed for demarcating thoughts is the challenge of reliably distinguishing large numbers of categories, and object categories alone may be insufficient for representing the complexity of a cognitive state.

What if, rather than tracking the rise and fall of particular object categories, we found a way to track semantic transitions more holistically? A growing body of research suggests evidence for the constructionist model of the mind: complex mental states emerge from flexible network-level interactions, and changes in active network configurations (i.e., time-varying functional connectivity) might signal boundaries between cognitive states[6–10]. However, current methods either require alignment of states across a group (e.g., by viewing the same movie stimulus)[11], referencing known states previously visited by an individual under stimulus control (a tiny subset of the range of possible states)[12,13], or focus on timescales too long to be relevant to measurement of single thoughts (with windows for representation of each cognitive state that approach a full minute long)[13]. Consequently, researchers investigating spontaneous thought have been unable to implicitly observe natural thought dynamics outside of stimulus control (e.g., using resting-state fMRI), where the timing and content of new thoughts is idiosyncratic. Arguably, implicit observation of this kind is central to any understanding of thought dynamics, as disrupting spontaneous thought for the purposes of explicitly communicating information about cognitive state itself disrupts the natural progression of states (analogous to the observer effect in physics).

Thus, we introduce a method to implicitly identify breaks between stable periods of brain network configuration (i.e., meta-state[13] transitions) at a single-TR timescale and using resting-state fMRI data from single participants.

Novel to our approach, we leverage similarities between event segmentation and spontaneous thought to present a preliminary psychological validation based on the correspondence of these transitions to known semantic and perceptual features. Neurocinematics studies have shown that well-made movies induce similar brain activity across participants in widespread low- to higher-order areas[14,15], and that some structures such as the hippocampus and angular gyrus are specifically tuned to event boundaries[11]. Thus, in addition to guiding brain state change-points, movies also exert control over the contents of cognitive states, and can be interpreted as a constrained analog to mind-wandering (where state transitions arise freely[7]). Separately, studies of spontaneous thought and event segmentation have also identified the same neural bases (e.g., hippocampus[16,17], angular gyrus[11,18], and precuneus and posterior cingulate[19–21]). We propose that a common theme of semantic integration[22] underlies both spontaneous thought and event segmentation, as both activities involve integrating new information with existing representations to shift the semantic focal point and move the storyline forward. Therefore, if transitions during movie-viewing reflect high-level semantic features (i.e., event boundaries) rather than low-level perceptual features, transitions at rest may also correspond to high-level semantic change (i.e., thoughts).

Upon demonstrating these properties, we generalize this interpretation by showing transitions to feature various similar neural properties when identified within unconstrained rs-fMRI data, as compared to the mv-fMRI data used in the analysis above. In parallel, we illustrate the continued psychological relevance of transitions identified in the rs-fMRI data by reporting correlations between trait neuroticism and our transition metrics in a manner predicted by recent personality science research on neuroticism, which characterizes one's proneness to negative thoughts and emotions. We focused on neuroticism because of mental noise theory, which proposes that trait neuroticism is linked to higher susceptibility to distractors and more lapses of attention; as a result, behavioral studies have found performance deficits on continuous tracking tasks and more variable reaction times within-participant across trials[23,24]. A related opinion in the realm of spontaneous thought suggests that high neuroticism is linked to excessive self-generated thoughts, supporting the theory of mental noise[25]. Further to supporting our validation of neural transitions as psychologically relevant during rs-fMRI, perspectives from public health suggest that a deeper understanding of neuroticism is likely to elucidate the mental and physical disorders linked to it[26].

## Results

**Detecting timepoints of interest.** We conducted our analysis on the 7 T Human Connectome Project dataset, which features movie-viewing fMRI (mv-fMRI) and resting-state fMRI (rs-fMRI) data gathered from 184 participants[27–29]. We converted each fMRI run into the expression of 15 known brain networks over time (Supplementary Fig. 1), then reduced its dimensionality from (15× time) to (2× time) using t-SNE[30,31]. In this reduced space, epochs with similar patterns of network activity fall in proximity. We hypothesized a pattern of spatiotemporal organization reflecting progression through a series of discrete thoughts, each centered around its own semantic focal point (e.g., what one will be having for dinner) serving as an attractor. Unstable network meta-states would yield dispersion in this space, whereas an attractor would cause points to cluster, yielding a worm-like series (arising from limited drift as thoughts evolve; Fig. 1a vs 1b).

Next, we identified changes in network activity by taking the squared Mahalanobis distance[32] between successive timepoints in t-SNE space for each fMRI run, obtaining a measure of meta-state change that we label a step distance vector. To stabilize the step distance vector, we repeated the dimensionality reduction and step distance vector creation process 100 times for each participant and each functional run. Peaks within the resulting mean step distance vector represent prominent reconfigurations of network meta-states, thus we called them network meta-state transitions (henceforth transitions). For purposes of baseline comparison, we also identified local minima in the mean step distance vector, which each represent a relatively stable network meta-state (henceforth meta-stable) (Fig. 2a).

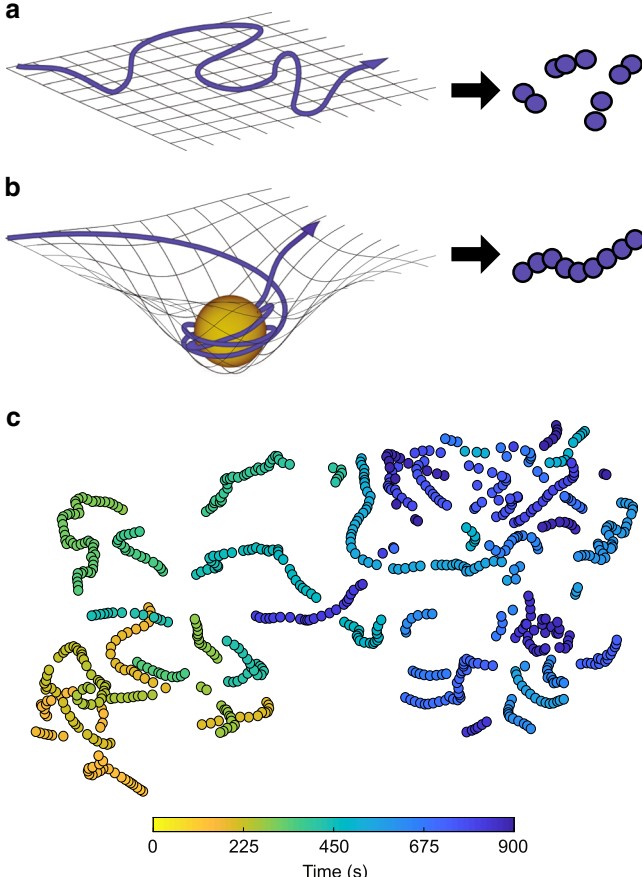

**Fig. 1 Network space representation. a** Continuous, random passage through the space of possible network configurations generates fragments in *t*-SNE space, as opposed to **b** contiguous, worm-like segments when an attractor holds network configurations in relative meta-stability. **c** An example reduced *t*-SNE representation involving both segment types, as observed from one participant's 15-min rs-fMRI scan. Here, the color gradient conveys progression through time from 0 to 900 s.

**Leveraging movie data to determine psychological meaning**. To assess whether these discovered moments of network reorganization held psychological relevance, we validated our approach using mv-fMRI data, examining their alignment to the onset of new semantic or perceptual movie features. As a starting point, visual inspection of the set of all participant transitions during movie-viewing revealed substantial alignment in transitions relative to rest (Fig. 2b, c). To quantify this, we obtained each participant's group alignment for each resting state and movie-viewing run, which describes the correlation between the individual's step distance vector and the corresponding median group step distance vector (i.e., conformity; see "Methods" section). We separated conformity values into runs of the same type, resulting in 723 movie conformity values and 722 rest conformity values. After feeding each set of values into a group bootstrap analysis, we found higher conformity for movie runs, mean $r = 0.27$, 95% CI: [0.26, 0.27], than for rest, mean $r = 0.04$, 95% CI: [0.03, 0.04], $r$ difference = 0.23, 95% CI: [0.22, 0.25]. This finding reflects past observations of film's unique ability to induce similar activity across participants in a wide variety of brain areas[11,12,33], and provides a preliminary link between transitions and naturalistic cognition.

As an alternate means of evaluating the influence of plot progression (i.e., progression of meaning) over transitions, we attempted to predict transition alignment based on the number of narrative events in each clip. Two expert raters came to a consensus on boundaries between events[34], which we defined as timepoints where a change in the movie triggers a new semantic focal point or evolves viewer understanding of the movie narrative. We correlated the number of these events in each clip with group alignment within each participant. This yielded 184 correlation coefficients that we entered into a group bootstrap analysis. Clips with more events per minute had higher group alignment, mean $r = 0.25$, 95% CI: [0.21, 0.29].

We also directly examined the temporal correspondence of transitions with other movie features within-participant. Consensus labels of sub-events (consisting of individual actions) and cuts were obtained from two expert raters. We also obtained lower-level feature timeseries for each clip that describe semantic, visual, and amplitude change, as well as change in head motion (see "Methods" section for a full description of the feature vector set). No other features were tested. Some features were correlated; for example, an event's end often coincides with a cut between shots, which in turn coincides with semantic and perceptual stimulus changes. These correlations could produce a result wherein transitions appear to reflect lower-level features, but only because they peak concurrently with the onset of new cuts or events. To disentangle high- and low-level feature contributions, we censored epochs of lower-level feature timeseries where higher-level event boundaries co-occurred (Fig. 3b). We reasoned that if lower-level features induce transitions, this should remain the case outside of censored epochs (see "Methods" section for further detail).

To determine the proportion of feature variance accounted for by transition versus meta-stable timepoints, we calculated eta-squared values for both uncensored and censored features after accounting for lag in the hemodynamic response function. Briefly, this involved computing the average level of each feature at each transition and meta-stable point found in the mv-fMRI data, working backwards in the feature vector based on the canonical hemodynamic response function (HRF)[35]. Higher eta-squared values reflect larger feature values at transitions versus meta-stable timepoint (i.e., alignment to transitions; see "Methods" section for full description of this calculation).

Transitions were strongly associated with movie features, with up to 60.8% of feature variance explained (event feature). The same alignment was not found when this analysis was applied to a noise data set consisting of the same participants, but with phase randomized data[36] (this noise generating procedure yields a data set with matched power spectrum to the original signal, but removes temporal structure). However, with the exception of sub-events, eta-squared values for all features decreased markedly after removing timepoints that could be confounded with event boundaries. Features corresponding to non-visual perceptual change, in particular, dropped to near-zero values (Fig. 3a). Thus, transitions showed a clear, albeit non-exclusive alignment to features pertaining to plot progression, with the top three predictors being event, sub-event, and semantic features.

**Extending from feature-rich movies to featureless rest**. We interpret this co-occurrence with various movie features as evidence that meta-state transitions correspond with psychologically meaningful mental events. However, to learn whether this information could be discovered within fMRI data was not our end goal, as this has been previously achieved using other methods (although our methods seem to substantially augment the fidelity of this mapping; see Supplementary Fig. 4). Instead, we regarded this as an important validation step needed to associate psychological meaning with our transition metric, such that the same metric could subsequently be used to infer similar

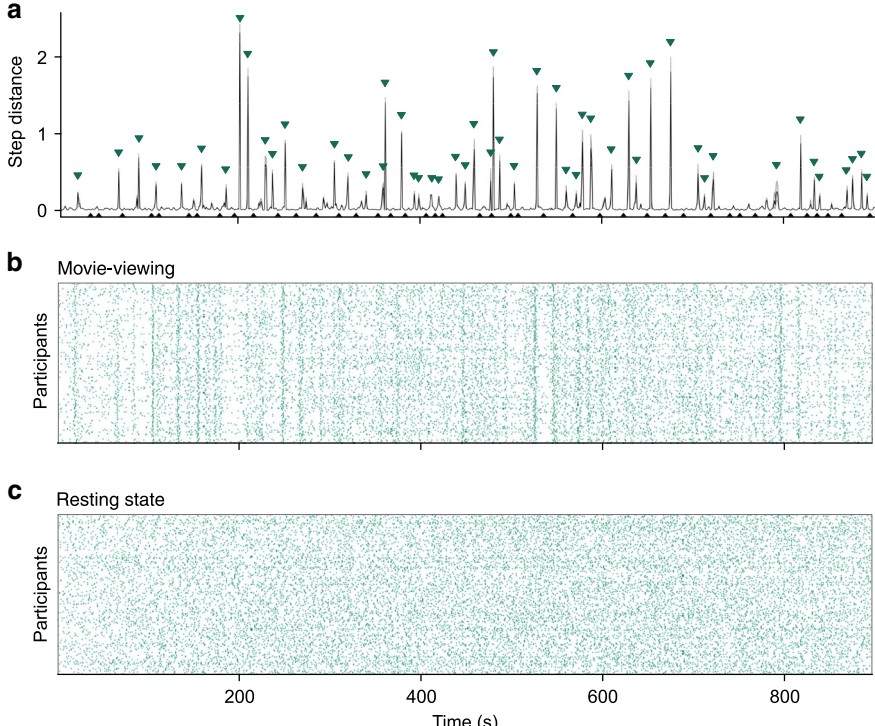

**Fig. 2 Identifying transition and meta-stable timepoints during movie-viewing and rest. a** A participant's mean step distance vector during one mv-fMRI run, with 95% percentile bootstrap confidence interval ribbon (largely imperceptible ribbon indicates stability over t-SNE iterations). Transition timepoints (green triangles) and meta-stable timepoints (black triangles) identified by a peak-finding algorithm. **b** All participants' transition timepoints for the same mv-fMRI run, with many peaks overlapping those of the example participant in **a**. **c** All participants' transitions for one rs-fMRI run. Alignment of peaks in **a** but not **c** reveals stimulus control over transitions.

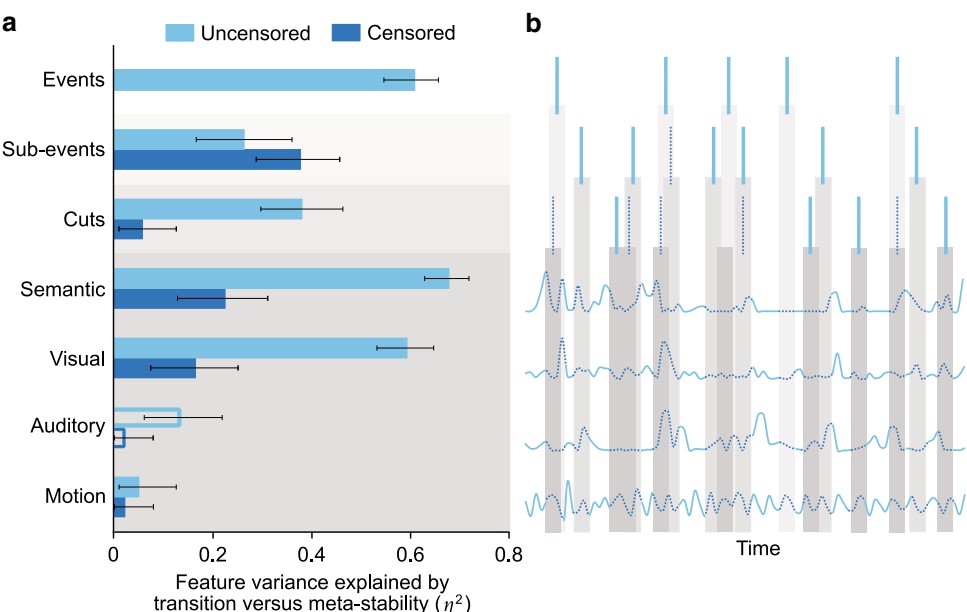

**Fig. 3 Variance in movie features explained using network meta-state transitions. a** Data are presented as eta-squared values describing the proportion of variance in movie features explained by alignment to transition vs. meta-stable timepoints. These values are calculated by: obtaining the mean transition-meta-stable feature value difference across all available mv-fMRI runs for each participant, using bootstrap procedures to obtain the mean difference across participants ($n = 184$), transforming bootstrap ratios and 95% percentile bootstrap confidence intervals into corresponding eta-squared values. Filled bars denote features that are aligned to transitions, whereas empty bars denote stronger alignment to meta-stability. **b** Diagram illustrating censorship of features for greater independence of test statistics. Blue lines describe activity in each feature across time. The cascading gray bars denote where lower-level features are censored by higher-level features.

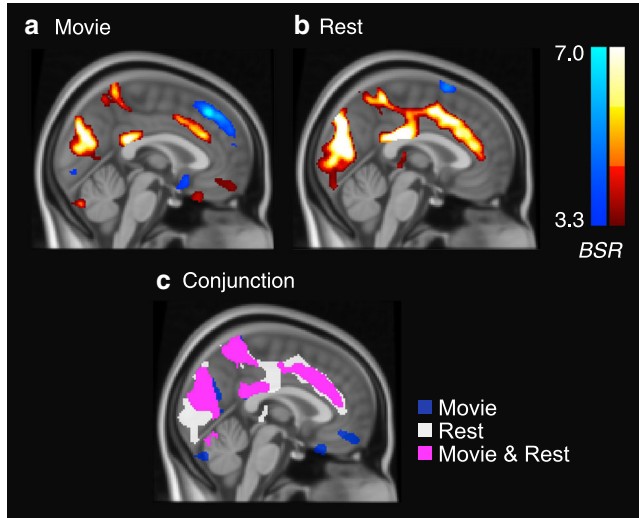

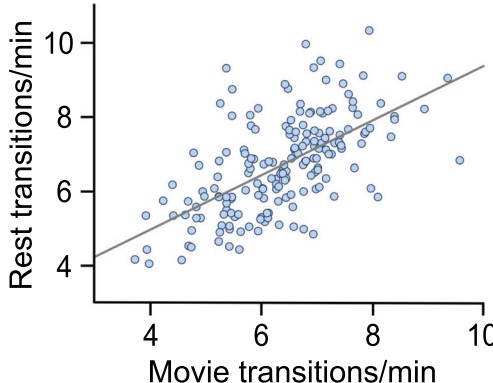

**Fig. 5 Transition structure generalizes from stimulus-driven to resting cognition.** Participants' movie-viewing transition rates correlated with their resting transition rates using a bootstrap correlation approach. Transition rate is calculated by taking the total number of transitions and dividing it by the time in minutes.

**Fig. 4 Spontaneous thought and attention regions distinguish transitions from meta-stability.** BOLD activity differences are quantified by the bootstrap ratio (BSR) statistic. The red scale represents transitions > meta-stability, whereas the blue scale represents meta-stability > transitions. **a** During movie-viewing fMRI, greater BOLD activity was observed during transitions than meta-stable timepoints, in several midline regions, including anterior cingulate cortex, posterior cingulate cortex, and visual association cortex. **b** During resting-state fMRI, greater BOLD activity was observed during transitions than meta-stable timepoints in similar midline regions. **c** Substantiating the subjective similarity of **a** and **b**, conjunction analysis revealed substantial voxel-wise overlap for regions activated during transitions (see Supplementary Tables 1–3 for voxel cluster details).

psychological meaning in new situations where insight is otherwise unavailable. As a test case, we targeted rs-fMRI data, which is very clearly associated with no stimulus, no shared structure across participants, and no shared structure within-participant. Because methods for indexing mental events in movies are reliant on such priors[11], whereas methods for indexing cognitive dynamics in rs-fMRI data[13,37] lack grounding in psychological phenomena, this reflected an initial attempt to evaluate the psychological dynamics of the resting mind.

To assess whether we could index mental events under these entirely uncontrolled conditions, we first interrogated possible similarities in the properties of transitions found in mv-fMRI and rs-fMRI data. First, we explored whether any consistent spatial pattern differentiated transition and meta-stable timepoints. Although transitions were defined on the basis of change in network activation, they captured both activation and deactivation of each network, so there was no circularity or bias toward the spatial characteristics of any one network.

We performed a conjunction analysis, searching for voxel activation clusters that independently distinguished transition and meta-stable timepoints for both mv-fMRI and rs-fMRI runs. We found that whether participants were engaged in movie-viewing or at rest, transitions were associated with very similar patterns of activation (Fig. 4a–c; Supplementary Table 3). In particular, transition timepoints were consistently associated with greater activation in several midline regions than meta-stable timepoints, including the anterior cingulate, posterior cingulate and precuneus. The insula is also powerfully and consistently implicated. By contrast, meta-stable activations were consistently found in dorsal and lateral frontal and parietal regions. In addition, mv-fMRI was associated with a variety of idiosyncratic meta-stable regions, including the temporal poles, ventromedial prefrontal cortex, both amygdalae, and part of visual association

cortex. This unique profile likely reflects the discrepant nature of engagement in movie-viewing (in which emotional, semantic, or visually salient aspects of the stimulus likely help sustain engagement in a particular state[38]), as opposed to rest.

We also evaluated whether an individual's transition rate generalized across task and rest. Before doing so, however, we wished to learn whether it was stable within task. To this end, we tested the internal stability of transition rate across the four rs-fMRI runs. Based on six pairwise correlations across the four resting-state runs, transition rate was highly consistent across runs, $ICC(2,k) = 0.79$, 95% CI: [0.74, 0.83], $F(183,549) = 4.8$, $p < 0.001$, revealing it to be a trait-like characteristic that can be adequately sampled using 15 min of rs-fMRI data. We then averaged participant-wise transition rates across rs-fMRI runs to stabilize the trait measurement, and regressed them against the average participant-wise transition rates computed from mv-fMRI runs. Transition rate at rest was correlated to transition rate during movie-viewing, $r = 0.60$, 95% CI: [0.51, 0.70] (Fig. 5), thereby further linking its properties across stimulus-driven and resting cognition.

**Link between trait neuroticism and transitions.** So far, the link we have drawn between transitions in rs-fMRI data and psychological meaning has been indirect, relying on the similarity of neural signals in rs-fMRI to those associated with stimulus features in mv-fMRI. To solidify this link, we aimed to link transition dynamics in rs-fMRI to psychological features. To do so, however, it was necessary to first establish whether transitions, like personality, featured trait-like characteristics that persisted over time. Fortunately, runs were sampled at four different timepoints, with runs 1 and 2 occurring in different scanner sessions in the same day, and runs 3 and 4 on another day. As a first step, we calculated the stability of transition rate within each day (i.e., pairwise correlations between runs that took place on the same day), and found that transition rate was highly consistent (day 1: $ICC(2,k) = 0.79$, 95% CI: [0.74, 0.83], $F(183,549) = 4.6$, $p < 0.001$; day 2: $ICC(2,k) = 0.80$, 95% CI: [0.75, 0.84], $F(183,549) = 5.2$, $p < 0.001$). Next, we calculated the average transition rate on each day 1 and day 2, and found that transition rate was still consistent, even across days, $r = 0.64$, $p < 0.001$, 95% CI: [0.54, 0.73]. Finally, we examined overall reliability across all 8 runs. We again found transition rate to be highly consistent, $ICC(2,k) = 0.86$, 95% CI: [0.84, 0.89], $F(183,1288) = 7.6$, $p < 0.001$. Together, these findings indicate that transition rate is stable across not only tasks, but across days, and may therefore be

regarded as at least a somewhat trait-like attribute that persists regardless of fluctuations in mood, fatigue, and other factors that fluctuate on a shorter timescale.

We next proceeded to link our data to neuroticism. Because our analyses above suggest that a large number of transitions correspond to a large number of thoughts, we interpreted transition rate as analogous to mentation rate and predicted that transition rate would be higher among individuals with high neuroticism. Consistent with this prediction, higher levels of trait neuroticism were associated with higher resting transition rates, $r = 0.15$, $p = 0.027$ (one-tailed), 95% CI: [0.00, 0.32], a result we confirmed in the larger 3 T data set, $r(970) = 0.09$, $p = 0.006$, 95% CI: [0.02, 0.15] (Supplementary Fig. 5). As a further test, we hypothesized that susceptibility to distraction would correspond to more idiosyncratic transitions during movie-viewing among high-neuroticism participants, yielding transition timing that conformed less to that of other participants (i.e., less alignment between an individual's stream of transitions and the group). Consistent with our hypothesis, higher trait neuroticism was associated with lower conformity during movie-viewing, $r = -0.18$, $p = 0.008$ (one-tailed), 95% CI: [−0.33, −0.06].

## Discussion

To summarize, our method characterizes a neurocognitive landmark based on transitions between brain network meta-states. Our approach is distinctive for its ability to observe fine-grained thought dynamics, and applicability to single individuals being scanned in any task context, with or without prior knowledge about the complexity of brain data under interrogation or the meta-states to be visited. As validation, we found transitions to be responsive to events in movie stimuli, and found features of transitions to be both trait-like and generalizable across task-based and task-free imaging contexts. Finally, we established construct validity of transitions by showing the frequency of their occurrence within resting-state fMRI data and lack of alignment to the group within movie-viewing fMRI data to predict higher trait neuroticism. Previous studies of neuroticism have measured mentation behaviorally and assessed neural bases by the size or hyperactivation of specific structures or overall functional connectivity between structures and/or networks; here, we provide neural measures of mental dynamics that bridge the gap between mentation and neural activity. Taking these observations together, we argue that neural meta-state transitions can serve as an implicit biological marker of new thoughts.

A thought is grounded in its contents. Therefore, our findings that meta-state transitions during movie-viewing best predicted event boundaries and onset of new semantic information support the interpretation that meta-state transitions align with the changes in semantic content across one's thoughts. These results complement those of Baldassano and colleagues[11], who used a Hidden Markov Model (HMM) approach on specific neural structures to find shifts in movie-viewing brain activity which co-occurred with event boundaries. Importantly, our method differs by its grounding in a specific and observable neural phenomenon (whole-brain network transitions), its applicability on an individual basis, and its capacity to detect transitions in uncharacterized task environments. This, in turn, makes our method widely applicable, being useful in situations where no priors or structure of any kind is available, including even resting cognition, as we have illustrated here.

We found that patterns of activation associated with transitions were mostly consistent across task contexts. We also found that the regions implicated have largely been previously associated with spontaneous thought. In particular, anterior cingulate and insular cortices are considered members of a salience network

that shifts attention to novel external and internal events[39], and the posterior cingulate cortex is a key node in the default mode network that is more active during task-unrelated than task-related thought[21,40]. By contrast, the only region consistently activated with meta-stability, angular gyrus, has been associated with sustained attention[41], a function that should be expected to operate in opposition to spontaneous switching to new thoughts. Beyond the angular gyrus, the profile of meta-stability diverged based on the nature of what participants were doing: in mv-fMRI, meta-stability was associated with activation in a variety of regions plausibly associated with the most engaging parts a movie stimulus (e.g., visual cortex, auditory cortex, fusiform face area, and amygdala, likely reflecting processing of visual and auditory sensations, faces, and emotions[42,43]). By contrast, in rs-fMRI, meta-stability was associated with regions plausibly associated with mentation in prefrontal cortex[44]. This overall pattern is consistent with the interpretation that whereas there are general neural mechanisms for transitioning to and locking onto a new state, brain activity during stable periods can be expected to reflect idiosyncratic thought contents.

Transition rate was correlated within runs of the same type, revealing it to be a trait-like characteristic that can be adequately sampled using 15 min of rs-fMRI data. This result builds on recent findings of trait-like properties of low-frequency chronnectome characteristics[37,45] by showing that higher-frequency network reconfigurations relevant to rapid, thought-like fluctuations in cognitive states are also trait-like. Furthermore, the average transition rate during movie-viewing was strongly related to average transition rate at rest. This relationship may be explained by the finding that participants segment commercial films and naturalistic clips with similar fractal structure[46], suggesting commonality between mentation during movie-viewing and real experiences. Film theorists lend further support to this finding, positing that narrative films and the techniques associated to creating them are not an attempt to reproduce reality, but are optimized based on real perception to control viewer's attention and mental state[47].

Our method yielded a neural measurement of mentation rate that we used to investigate how neuroticism relates to thought dynamics. In particular, the mental noise hypothesis proposes that individuals with high neuroticism are more susceptible to intrusive thoughts and distraction by irrelevant information[23–25], presumably yielding more frequent and less predictable changes in cognitive state. Consistent with this proposal, we found trait neuroticism to predict higher transition rates (i.e., mentation rate) during rest, as well as lower conformity of transition temporal structure to that of the group during movie-viewing. We interpret these two observations as reflecting increased mental noise in individuals with high trait neuroticism, which in turn supports the construct validity of transitions as a measurement of thought dynamics in both rs- and mv-fMRI. It also serves as a first neural confirmation that neuroticism indeed entails a noisier mind.

The current results invite comparison against related approaches, such as the application of HMM to movie data. Although the current approach outperforms HMM in detection of movie features (Supplementary Fig. 4), it is more informative to focus on their qualitative differences. In particular, we suggest that HMM is a better choice for testing hypotheses where relevant states can be learned from other data. For example, group movie-viewing fMRI data can be used to identify states that individuals watching the movie are likely to visit, and then measure whether each individual visits them[11]. Similarly, an individual's encoding run can be used to test whether states participant visited are later reinstated during free recall[12], and the tendency of an individual to visit specific, pre-defined states while at rest can be measured[12]. By contrast, the current approach is agnostic to which specific

states are visited. However, it is sensitive to displacement within the full space of possible states. Its main output is the times at which state transitions of any kind occur, whether or not we have any idea what the states may look like. This means the technique can be applied effectively to a first pilot participant's very first fMRI run. It is therefore ideal for finding temporal structure in uncharacterized task environments, such as resting cognition, where participants engaged in free thought are likely to navigate a vast space of possible states, mostly unpredictably.

By lending a level of validity and reliability to measuring thought dynamics that were unavailable using past introspective approaches, our approach also creates opportunities to understand cognition. Although our analysis has focused on movie and resting fMRI data from healthy adults, our methods are applicable to a wide range of tasks and populations, as well as even case studies, as no group data are required. To illustrate some unique affordances, one can ask other individual-difference questions such as: does transition rate influence a person's ability to remain engaged in sustained attention task? Or, using a more task-based approach: while it is known that novel stimuli are initially attention-grabbing, are their differences in the thought dynamics associated with watching a favorite movie for the first time, relative to the fifth time? Regarding special populations, can measures of thought dynamics serve a clinical function by offering early detection of disordered thought in schizophrenia, or rapid thought in ADHD or mania?

In closing, it is interesting to consider one further example application that can be addressed with the information already at hand: how many thoughts do we experience each waking day? Extrapolating from our observed median transition rate across movie-viewing and rest of about 6.5 transitions/min, and a recommended sleep time of 8 h, one could estimate over six thousand daily thoughts for healthy adults of a young-adult demographic similar to the one used in our analysis. Although further interrogation of meta-state transitions would be needed to employ such a measure with confidence, availability of a tentative answer further highlights how the current approach may be fruitful in advancing how we think about thought.

## Methods

**Human Connectome Project dataset.** Neuroanatomical and functional data were collected by the WU-Minn Human Connectome Project (HCP) consortium[28]. In a prior analysis, a 3 T resting-state fMRI (rs-fMRI) data set of 1003 participants (age $M = 28.7$ years, SD = 3.7 years; 534 female) was used by the HCP group to generate spatial maps of typical brain networks that may be found in rs-fMRI participants through a process involving group-PCA and group-ICA[48,49] Our analyses involved applying these maps to the 7 T HCP data set that contained both mv-fMRI and rs-fMRI scans, and that was the subject of all of our own analyses. Although this 7 T data set is described elsewhere[27–29,50], briefly, it consists of fMRI scans from 184 participants (age $M = 29.4$ years, SD = 3.4 years; 112 female). Each participant underwent four 15-min mv-fMRI and four 15-min rs-fMRI runs; functional images were acquired using a multiband gradient echo-planar imaging (EPI) pulse sequence (TR 1000 ms, TE 22.2 ms, flip angle 45°, multiband factor 5, whole-brain coverage 85 slices of 1.6 mm thickness, in-plane resolution $1.6 \times 1.6$ mm$^2$, FOV $208 \times 208$ mm$^2$)[51–54]. During each movie run, participants watched three or four movie clips interspersed with 20-s rest periods as well as an 84-s validation clip repeated at the end of each run (due to its repetition, we did not include this clip in our analyses).

In addition, high-resolution T1-weighted and T2-weighted scans were gathered (TR 2400 ms and 3200 ms, TE 2.14 ms and 565 ms, flip angle 8° and variable, 0.7 mm thickness, in-plane resolution $0.7 \times 0.7$ mm$^2$, FOV $224 \times 224$ mm$^2$) for purposes of group anatomical alignment. Data collection was approved by the Washington University institutional review board[27] and performed by the HCP consortium, which also gathered informed consent from all participants at the time of data acquisition. Access to these data sets was granted by the HCP consortium, and acknowledged by the Health Sciences Research Ethics Board at Queen's University. No participants were excluded from analysis.

**Transforming functional data to 15-network representation.** We mapped the 15 spatial maps resulting from the 3 T resting-state group-ICA decomposition onto each 7 T participant's resting state and movie-viewing data using FSL's dual

regression function[48,49], a method in which known spatial configurations are regressed against new data to transform 4D functional data into a set of timeseries (one per spatial map, Supplementary Fig. 1a).

Although larger network set sizes were available from the HCP group (ranging from 15- to 300-brain-network solutions), we observed that set size had little material impact on trajectory estimates, and therefore selected the simplest available (15-network) solution. To increase signal-to-noise ratio and to dampen short timescale network fluctuations, we temporally smoothed each resulting timeseries with a moving average filter (span = 5 s). This procedure yielded a smoothed timeseries for each brain network reflecting that network's activation over time. We combined these timeseries to create separate (time × network) matrices for all four mv-fMRI and all four rs-fMRI runs for each participant (Supplementary Fig. 1b).

**From network representation to network meta-states.** In preparation for using the Mahalanobis distance metric on the data, we applied the $t$-distributed stochastic neighbor embedding ($t$-SNE) algorithm to reduce the dimensionality of each matrix from 15 dimensions to 2 dimensions at the default perplexity setting of 30[30] (Fig. 1c; Supplementary Fig. 1c). We found the perplexity setting had little impact on our analysis, and therefore selected what is regarded as a moderate (and default) value. We define the reduced space as the meta-state space as each two-dimensional timepoint is a higher-order (i.e., meta) representation of a 15-network activity configuration, as in the method by Miller and colleagues[13].

Notably, in Miller and colleagues' approach[13], they created a low-dimensional (higher-order) representation by first defining the space of possible meta-states as a discrete 5-dimensional state space, with each dimension representing a distinct group temporal ICA component derived from participants' functional data (i.e., connectivity patterns). Whole-brain activity was expressed as a weighted combination of these components over time. To map each timepoint onto their meta-state space, they then discretized each weight at each timepoint according to its signed quartile. In contrast, our approach relies on dimensionality reduction algorithms to discover changes in meta-state directly from the continuous-valued 15-brain-network representation. We selected this approach because it affords flexibility in the designation of each meta-state by mapping each one onto a continuous two-dimensional space instead of a discrete 5-dimensional state space. Just as the Greek philosopher Heraclitus noted, "No man ever steps in the same river twice, for it's not the same river and he's not the same man", our approach is aligned to the very likely possibility that meta-states are continually evolving. Our approach also differs by drawing on published reference networks derived from a static group of 1003 participants (i.e., the 3 T data set described above[28,48,49]), rather than a set of networks derived from the specific data set under interrogation.

**Detecting network meta-state transitions.** To derive from our $t$-SNE representation a global measure sensitive to changes in network meta-state, we computed the Mahalanobis distance in position within this low-dimensionality $t$-SNE space across subsequent timepoints. Covariance matrices were empirical (i.e., calculated based on the input sample). The resulting step distance vector for each fMRI run of each participant should peak at points in the timeseries where shifts in network meta-state occur. To address potentially divergent results across repeated $t$-SNE algorithm runs, we repeated the dimensionality reduction and step distance vector creation process 100 times for each participant and each functional run. Then, we took the mean across the 100 step distance vectors for that run (Fig. 2a). However, even 95% confidence intervals were tightly constrained.

We applied a peak-finding algorithm on each mean step distance vector to identify transition timepoints at which the step distance satisfied a minimum peak prominence threshold of 0.06, the value at which ~80% of all step distance values fell under the 5th percentile transition-associated step distance value. Setting a prominence value rather than applying a high pass filter allows the algorithm to consider step distances in the neighborhood surrounding the peak being evaluated and results in more robust transition selection. To find meta-stable timepoints, we inverted the signal and specified a minimum peak width of 10; this parameter ensured that timepoints would only be identified within persistently meta-stable periods. One example of a participant's identified transitions and meta-stable timepoints are shown in Fig. 2a with green and black triangles, respectively. Under these parameters, the median step distance value associated with transition timepoints was 0.48 (5th and 95th percentile: [0.09, 1.79]). The median exceeds 94% of all values, whereas the lower bound exceeds about 80% of all values. By contrast, the median step distance associated with meta-stable timepoints was 0.02 (5th and 95th percentile: [0.01, 0.03]), falling below about 94% of all values.

**Transition characteristics in simulated realistic fMRI data.** We used fmrisim from the BrainIAK[55] toolbox to generate a noise data set consisting of phase randomized participant data for the first movie run. Phase shifts were carried out voxel-wise for each participant (i.e., random different phase shifts instead of the same across all voxels). After transforming the fMRI signal into the network representation through dual regression of the 15-network spatial ICA maps, we followed the method outlined in our paper (i.e., ran 100 iterations of the $t$-SNE algorithm, calculated Mahalanobis distances). The final step distance vector for

each noise participant consisted of the mean step distance vector across these 100 iterations.

Inspecting the individual $t$-SNE iterations on the noise data, we notice that results often consist of a few lengthy contiguous clusters (i.e., few smaller transitions), in contrast to the several contiguous clusters we see in the $t$-SNE projections of real data that result in numerous identified transitions. Furthermore, comparing a participant's mean noise step distance vector against their real mean step distance vector, we notice that the real data yields transitions that are more consistent across repeated $t$-SNE algorithm runs (Supplementary Fig. 2). These results highlight the importance of carrying out the repeated $t$-SNE algorithms step to stabilize the transitions that are subsequently identified.

**Movie feature vectors**. To link our identified neural transitions to participants' experience, we investigated how transitions mapped onto well-characterized movie features (events, sub-events, cuts, semantic, visual, and auditory), as well as each participant's recorded head motion during the scan. Sub-events describe directly what an actor or multiple actors are doing or saying ("he is walking down the stairs", "they are riding in a car"), or they describe the motion of important objects ("a meteorite strikes the ground", "a car drives down the street"). We defined an event as a meaningful cluster of sub-events that describes a larger, overarching goal achieved by the sum of its parts (i.e., fine-grained and coarse-grained events[34]). Cuts referred to boundaries between two separate camera shots. Two raters used a video coding tool (Datavyu[56]) to independently identify boundaries demarcating events and sub-events, then met to discuss any differences in their ratings and achieve consensus event boundary timepoints. Using this approach, raters created consensus event and sub-event boundary segmentations for all four movie runs (14 clips). Only one rater logged cuts, as the objective nature of camera positions left little to discussion. Across the 14 clips found during the four movie runs, raters identified an average of 7.5 events, 37.3 sub-events, and 46.2 cuts per clip. A binarized timeseries was created for each logged feature, with onsets allocated to the nearest 1 s time bin (corresponding to the 1000 ms TR during which each fMRI volume was gathered).

The HCP group supplied two types of feature labels for the movie stimuli: semantic-category labels that described the high-level semantic features contained in each 1-s epoch of the film[57], and motion-energy labels that described low-level structural features in the same epochs[58]. There were 859 semantic features and 2031 motion-energy channels that expressed changes in the semantic content and motion-energy of each epoch, respectively. By summing across all semantic features and taking its derivative, we obtained a measure of overall magnitude of change in semantic content at each epoch. Similarly, we summed across all motion-energy channels and took its derivative to obtain a measure of overall magnitude of change in perceptual features at each epoch. We also took the absolute value of the auditory amplitude vector derivative for each movie run as a measure of magnitude of change in volume. Finally, to rule out the possibility that transitions are a motion artifact, we obtained, for each participant, relative root-mean-square (RMS) change in head position[59], corresponding to a vector of head movement over time.

**Disentangling event boundaries and movie features**. Whereas the feature vectors obtained above were correlated, we derived from them a set of independent feature vectors by censoring epochs where event and sub-event boundaries, as well as cuts were present. This was done by dropping values in a 3-s window around feature boundaries and ensured that apparent effects in lower-level features would not be explained by correlation to higher-level features. Event boundaries censored all other vectors; sub-event boundaries censored all vectors other than events; and cuts censored all vectors other than events and sub-events (Fig. 3b). To prevent spurious effects related to clip onset within movie runs, we also censored the first 6 s of each clip for both the uncensored and censored feature vectors.

**Movie and movement features at transition vs meta-stability**. For each transition and meta-stable point found in the mv-fMRI data, we next computed the average level of each feature accounting for hemodynamic response function (HRF) lag (working backwards based on the canonical HRF[35], sampling from a feature window 3–6 s prior to each transition and meta-stable point). We then averaged across all onsets of the same type for each participant. Thus, each participant ultimately had two values for each feature vector: one representing the average feature vector value at a transition timepoint, and an analogous value at a meta-stable timepoint.

We ran a $t$-test comparing these two values across participants for each feature, and used each resulting $t$-statistic to compute the proportion of variance that was explained in the feature vector (i.e., eta-squared) by the presence of a transition. Next, we used a non-parametric bootstrapping analysis to obtain a 95% CI for each feature[60]. Using participants' transition-baseline feature value difference as input data, this approach constructs a sampling distribution of the mean by resampling 1000 times with replacement across participants. Then, the $t$-statistics corresponding to the upper and lower bounds of each CI were converted to eta-squared values (Fig. 3a).

**Comparison of real versus simulated data**. We also used the phase randomized noise data set described previously to conduct a secondary analysis investigating

the association between noise transitions and movie features. These results provide a baseline comparison to the real eta-squared values we obtained, which describe the alignment of transitions to features. We found no significant association between transitions and movie features.

**Performance of alternative embedding approaches**. Having identified strong prediction of various features, we next sought to determine how crucial our specific embedding approach (i.e., $t$-SNE) was to identifying transitions that mapped strongly onto movie features. To this end, we derived five additional sets of transition and meta-stable timepoints: one using the unreduced (time × network) representations of each participant's mv-fMRI data (Fig. 1b), a second and third set using a 2-dimensional representation of their mv-fMRI data obtained through principal components analysis (PCA) and independent components analysis (FastICA), a fourth using the method described by Miller and colleagues[13], and a fifth using the Hidden Markov Model (HMM) approach described by Baldassano and colleagues[16].

For Miller and colleagues' method, we began by regressing the spatial ICA maps corresponding to the 50-network decomposition into the functional data to obtain a 50-network representation of brain activity over time and match the dimensionality of their initial input data. Then, we calculated the pairwise correlations between 44 s time-window of activity in each network. This process was repeated for the entire timeseries by sliding the time-window in increments of 1 s. As a result, a given participant's brain activity at a time-window was expressed as the set of pairwise correlations between each of the networks (i.e., a connectivity pattern). Then, we applied FastICA to obtain a 5-dimensional representation of activity over time, with each of the 5 components representing a particular connectivity pattern. After discretizing the weights of each component at each time-window according to its signed quartile, the resulting 5-dimensional representations were used to derive corresponding step distance vectors, as well as transition and meta-stable timepoints.

Specifically, for each embedding (15-dimensional for the unreduced timeseries, 2-dimensional for the PCA and ICA approaches, and 5-dimensional for Miller and colleagues' method), we calculated the Mahalanobis distance across subsequent timepoints and applied the peak-finding algorithm to identify transition timepoints; the minimum peak prominence threshold selection strategy and peak-finding parameters for meta-stable timepoints remained unchanged.

For the HMM approach, we fit the HMM to the 15-network representation of brain activity, which segments the data into the number of events expected from human-rater event segmentations of the movie stimuli. This step produces a probability matrix with a row for each timepoint and a column for each event. The entries of this matrix reflect the probability of a certain timepoint belonging to that event. To pinpoint the transition between event $k - 1$ and $k$, we identified the cross-over timepoint at which the probability of event $k$ surpasses the probability of event $k - 1$. By contrast, we deemed the timepoint at which the maximum probability value was observed within event $k$ (i.e., the maximum within its corresponding column) to be meta-stable.

As above, we used these five additional sets of transition and meta-stable timepoints to obtain the proportion of variance that was explained in the uncensored feature vectors by the presence of a transition for each embedding approach (Fig. 3b). Transitions derived directly from the 15-dimensional (time × network) representations predicted a moderate proportion of variance in events, but not for lower-level event-based features such as sub-events or cuts. PCA, ICA, Miller's method, and HMM-based transitions performed worse than $t$-SNE transitions for all feature categories. We interpret the strong performance of $t$-SNE in predicting features (especially semantic features) as reflecting its unique ability to distill important local and global structure from the data.

Consistent with this idea, as dimensionality increases, distance metrics are understood to lose their usefulness as the distances to the nearest and furthest point from any reference point approach equality[61]. Zimek and colleagues[62] further showed that if the dimensions are correlated rather than independent and identically distributed, then considering subsets of dimensions can improve the performance of distance metrics such as Euclidean distance. This likely explains the poor unreduced feature prediction in our analysis, and supports the usage of a dimensionality reduction algorithm, as the spatial networks in the 15-dimensional (time × network) representations that we submitted to analysis were not independent from each other. Substituting Mahalanobis for Euclidean distance can additionally account for residual correlations between reduced dimensions[32].

In spite of this, in our analysis, poor PCA and ICA performance signaled that the kind of dimensionality reduction approach also matters. PCA and ICA are linear methods that seek to preserve global structure, whereas $t$-SNE can balance a trade-off between local and global structure. Upon exploring the variable loadings for PCA results, we observed that the two principal components were weighted heavily toward visual networks that likely explained substantial global variance, but at the cost of sensitivity to changes in networks related to higher-level conceptual processing. As a result, PCA-based transitions performed comparably to unreduced transitions in the visual category and correlated categories (e.g., auditory), but worse in semantic categories. The increased restrictiveness that results from the additional requirement of statistical independence imposed for ICA components may explain why it performs even worse than PCA.

Transitions derived from Miller and colleagues' method were not associated with feature changes. As we are looking for moment-to-moment changes in network reconfiguration, one possible explanation is that the size of the time-window used to correlate across the 50 networks (44 s) was too long to be sensitive to feature-related changes at a certain timepoint. Thus, we tested the smallest time-window that could be evaluated (3 s), but still found that no features were associated with transition relative to meta-stable timepoints (all $ps > 0.28$). A 2 s time-window is impossible to evaluate as the pairwise correlation values are −1 or +1, reflecting the linear relationship between two points. Furthermore, reducing dimensionality by distilling the 50-network pairwise correlations to 5 connectivity patterns may not provide enough flexibility to capture the different mental states linked to movie-viewing.

Of the embedding methods we tested, HMM-based transitions were second only to our $t$-SNE-based approach at aligning to feature changes and predicted a moderate proportion of variance across all but auditory features. In addition, an important limitation, particular to this method, was that the number of transitions to be placed with each fMRI run had to be pre-defined. This can be accomplished in two ways: on the one hand, one can use information about the stimulus to define the expected number of events (as we have done), although this arguably constitutes a form of peeking in analyses such as the current one where the goal is to recover stimulus features. Furthermore, this strategy is also impossible where no stimulus-based guidance exists (e.g., during resting-state fMRI). Another way of pre-defining the number of transitions is to perform an optimization at the group-level. However, this approach assumes that the pre-defined number of events is present in each participant's data, whereas there may be individual differences in the number of event boundaries. Outside of constrained situations where participants are all viewing the same stimulus, there may be little shared structure in the type or timing of states across participants (also as in rs-fMRI). These considerations limit the application of HMM-based methods to situations where priors exist (such as a known stimulus, shared group structure, or prior exposure). By contrast, our goal was to only initially validate our technique in an environment where the stimulus could be well-characterized, with the intention of subsequently applying it largely in situations where no priors are available.

These points made, as a further validation step it was helpful to consider that, to the extent our $t$-SNE based and the previously-validated HMM-based approach are both successful at identifying cognitively meaningful event boundaries, the transitions we identify using the $t$-SNE approach should have some alignment to HMM-based transitions. To that end, we calculated the proportion of HMM-based transitions that were found within a 3-s window of a $t$-SNE-based transitions for each participant and movie run. On average, 40% of HMM-based transitions were found in $t$-SNE-based transitions, confirming that although we were not able to achieve the same level of feature-detection performance with HMM, it nonetheless indexed a similar neurocognitive construct. The lower percentage may result from forcing the placement of transitions in the HMM-based approach, thereby identifying transition timepoints that may not correspond with actual mental state change.

**Mapping between network node activity and movie features**. We regressed each movie feature timeseries on each participant's 15-network node timeseries to investigate the predictive power of individual networks. This yielded 184 adjusted $R$-squared values for each of the 6 features. The highest mean adjusted $R$-squared value was for the semantic feature (mean $R^2 = 0.0033$), signaling that individual networks were not meaningful predictors of any feature.

**Mapping between network node activity and transitions**. We imported the transitions and meta-stable timepoints identified at rest and relative head motion from the rest runs. Next, we used the same strategy as with the movie feature analysis. In particular, we reasoned that if participants' head motion causes transitions, then the average value of relative head motion at a transition timepoint should be significantly larger than the average value at a meta-stable timepoint. We obtained the value representing this transition-meta-stable head motion difference for each participant, then fed these values into a bootstrap analysis comparing against the null hypothesis of zero. We found no significant association between transitions and head motion at rest, mean transition-meta-stable step distance difference = −0.00, BSR = −1.10, $p = 0.273$, 95% CI: [−0.00, 0.00].

**Event structure influence on transition group alignment**. Visual inspection revealed considerable structure in transitions across participants during movie-viewing (Fig. 2b), complementing prior findings of local coordination of brain activity as participants watch well-made films. To formalize this observation, we tested for higher group alignment in movie runs than rest runs. For each participant and each run, we took the Fisher transformation of the correlation between the log of their step distance vector and the log of the median group signal (excluding the step distance vector of the participant in question). To avoid potential effects resulting from the onset and offset of the run, we excluded the first and last five epochs of the step distance vectors. Then, we compared all group alignment values from the four mv-fMRI runs against zero. The same boot-strapping analysis was carried out for all group alignment values for the four rs-fMRI runs.

To provide an alternate way of testing the influence of narrative events over transitions, we correlated the number of events in each movie clip with the degree of group alignment of transitions for that clip. For each clip, we divided the number of events in that clip by its duration in minutes. For each participant, we obtained their clip conformity by taking the Fisher transformation of the correlation between the log of their step distance vector and the log of the median group signal (excluding the step distance vector of the participant in question). This was repeated for each clip within a movie run rather than the movie run in its entirety; thus, each participant had a vector describing their clip conformity, wherein each element corresponded to a conformity value for a particular clip. We calculated the Pearson correlation between the events per minute and conformity vector within each participant, resulting in a distribution of 184 correlation coefficients. We fed this distribution into a bootstrapping analysis (as described previously when we used this strategy to obtain 95% confidence intervals) with 1000 samples to test whether the correlation between events per minute and group alignment was significantly different from 0.

**Effects of peak threshold and temporal smoothing parameters**. To evaluate the importance of these parameter settings, we tested the effects of varying minimum peak prominence (MPP) values and temporal smoothing spans on the movie feature analysis. We derived sets of transition and meta-stable timepoints from 11 MPP values: [0.01, 0.02, …, 0.11]. These timepoints were fed into the movie feature analysis. Results are shown in Supplementary Fig. 3a, with results of the original, moderate MPP value (0.06) represented by the middle bar in each 11-bar cluster. As we aimed to identify neural transitions that corresponded with a new thought, which we proposed are analogous to event boundaries and semantic content changes in movies, we are reassured that the alignment between transitions and events remained stable across MPP values. Furthermore, in the semantic feature, alignment strength increases as values approach the actual MPP value (0.06), but then remain stable for values greater than the actual MPP value.

Smoothing occurs on the 15-network activity configuration, just before this representation is used as input into the dimensionality reduction algorithm. We originally used a 5-s span, as this is close to the time-to-peak in the hemodynamic response function. However, to probe this parameter, we derived meta-state space representations for data smoothed with the following spans: [1 s, 3 s, 5 s, 7 s, 9 s]. From these five sets of data, we derived five sets of transition and meta-stable timepoints. As above, these timepoints were fed into the movie feature analysis. Results are shown in Supplementary Fig. 3b, with results of the original span (span = 5 s) represented by the middle bar in each 5-bar cluster.

We also determined transition rate for each of these five spans. A one-way ANOVA revealed that there was a significant difference in mean transition rate between the spans. Our chosen span was also often (but not always) optimal from a signal detection perspective. However, follow-up pairwise $t$-tests between our chosen span (5 s) and each of the other spans revealed no significant differences (all $ps > 0.27$).

In conclusion, an exploration of the parameter space confirmed that our selection of this smoothing window was desirable in terms of the alignment of our measures to features, yet did not elicit a reliable difference on the analysis outcome relative to no temporal smoothing (i.e., span = 1 s).

**Spatial correlates of transitions vs meta-stable timepoints**. Using the transition and baseline onsets, we next performed a voxel-wise conjunction analysis in which we sought to identify stable spatial correlates of network meta-state transitions that could be found across rest and movie runs. Through this analysis, we wished to learn whether any set of transition predictors could link brain activity for which we have insight into psychological relevance (mv-fMRI) with brain activity for which we do not (rs-fMRI). To this end, we sampled the average fMRI image at transition and baseline onsets for each participant in the same manner as with feature vectors above, but without correcting for HRF lag (as the predictor and dependent variables were affected by the same delay). In this case, however, we created participant-wise transition and meta-stable timepoint averages not only for mv-fMRI runs, but also (separately) for rs-fMRI runs. We also spatially smoothed each image using a 6 mm FWHM Gaussian kernel (at the lower bound of optimal parameters for overcoming inter-subject variability[63]).

**Correlation between personality and mental dynamics measures**. The HCP administered the Neuroticism/Extroversion/Openness Five Factor Inventory (NEO-FFI[64]). We used participants' scores for each facet of human personality (neuroticism, extroversion/introversion, agreeableness, openness, and con-scientiousness) as correlates to average rs-fMRI transition rate and conformity (specific to mv-fMRI). Although we had a directional hypothesis for the relationship between neuroticism and our two measures of mental dynamics, correlations between other traits and measures of mental dynamics were considered exploratory. We used a bootstrap correlation approach with 1000 samples to assess correlation between neuroticism, transition rate, and conformity.

**Head motion and transitions at rest**. Previously, we showed that there was no association between head motion and transitions during movie-viewing. To ensure that this was also the case in our resting-state data, given that motion can be higher

during resting-state scans[65], we repeated the same feature analysis strategy for head motion and transitions at rest and found no difference between average head motion during a transition versus at rest, $BSR = -1.10$, $p = 0.273$, 95% CI: [−0.00, 0.00].

**Correlations of other personality traits to transition rate**. For clarity and completeness, we report results describing the relationship between transition rate and each of the personality traits (Supplementary Fig. 6). To maximize statistical power, we conducted this analysis using the large 3 T data set. To address potential interrelations among the traits, we isolated each trait by controlling for all other traits in the descriptive analysis. In so doing, we observed that openness was negatively associated with transition rate, possibly reflecting the relationship that has been demonstrated between this trait and transliminality[66], defined as a "tendency for psychological material to cross thresholds into or out of consciousness"[67]. Based on our definition of a thought as containing transient cognitive state, more material crossing into consciousness should correspond to a higher volume of thoughts. Based on the ideas presented here, this, in turn, should lead to a higher transition rate. We should caution, however, that our interpretation is speculative and post hoc. We therefore suggest this observation is best regarded as evidence to support hypothesis generation for future research.

**Impact of sex differences in neuroticism on transition rate**. Because it is well established that across cultural contexts, neuroticism tends to be higher in females than males[68], we organized the 7 T data set by sex, and evaluated the difference in the slope of the correlations established for males versus females (no participants were marked as "other"). We found that neuroticism was associated with higher transition rate in females than in males, $r_{diff} = 0.28$, 95% CI: [0.00, 0.57]. However, this pattern did not survive translation to the 3 T data set, $r_{diff} = 0.08$, 95% CI: [−0.05, −0.21], providing mixed evidence for the sex specificity of our findings.

**Effects of clip emotionality on mental dynamics measures**. Perception is biased toward arousing stimuli[38]. An emotionally arousing film stimulus should, therefore, increase the amount of control exerted by the stimulus over participants' perceptions and, in so doing, decrease the rate of spontaneous, stimulus-independent cognitions. We operationalized this hypothesis as follows: clips with higher arousal ratings will be associated with lower transition rates and greater alignment of transitions across participants.

To establish arousal ratings for the film clips, we gathered responses from 12 first year psychology students who watched, in a random sequence, the 14 movie clips that were presented during the mv-fMRI runs. After viewing each film clip, participants evaluated the arousal of the clip. We used these ratings to establish an average arousal rating for each film clip. This resulted in a vector of 14 average arousal scores.

Returning to the brain data, we constructed a parallel 14-clip transition rate vector for each mv-fMRI participant. We then calculated the correlation between the arousal vector and each participant's transition rate vector. This resulted in a correlation coefficient for each participant that described the relationship between the level of arousal associated with each clip, and the participants' resulting transition rate. Inputting this distribution of coefficients into a bootstrap analysis, we observed that higher arousal was reliably associated with fewer transitions, mean $r = -0.16$, $BSR = -8.83$, $p < 0.001$.

We also obtained a 14-clip conformity vector describing the degree of similarity between the participant's step distance vector and the group for each clip. We calculated the correlation between the arousal vector and each participant's conformity vector. The distribution of correlation values describing the relationship between arousal and conformity was fed into a bootstrap analysis. We found that higher arousal was associated with more conformity, mean $r = 0.10$, $BSR = 4.73$, $p < 0.001$.

Taken together, our analysis confirmed our hypothesis that more emotional stimuli would better capture and maintain the viewer's attention, resulting in fewer overall transitions (i.e., lower transition rate) and higher similarity to the group (i.e., conformity). Intriguingly, this interpretation is also consistent with our observation that the amygdala is associated with meta-stability during movie-viewing.

**Bootstrapped confidence intervals**. We used a bootstrapping approach to determine confidence intervals, descriptive statistics, and for statistical evaluations throughout the analysis.

We used a bootstrap approach to determine the stability of the step distance vectors across multiple $t$-SNE algorithm runs and to find 95% confidence intervals for eta-squared values in the psychological relevance analysis.

For the step distance vectors, we fed the step distance vectors resulting from all 100 iterations into a bootstrapping analysis. Then, for each timepoint, the 100 values from each iteration were resampled with replacement and the mean of that sample recorded. This resampling process was repeated 1000 times to build a sampling distribution of the mean for each timepoint. Confidence intervals around each element of the participant's mean step distance vector are derived from the 5th and 95th percentile values of the corresponding sampling distribution.

For the eta-squared values, we first obtained a vector for each feature describing the difference between average transition and meta-stable timepoint feature values

within-participant (i.e., 184 elements representing the difference for each participant). The transition-meta-stable difference vectors for each feature were fed into the bootstrap analysis to obtain 95% confidence intervals around the mean difference. Then, these bounds replaced the mean in the eta-squared calculations to obtain corresponding eta-squared values at the bounds of the confidence interval.

**Bootstrapped descriptive statistics**. We followed a similar procedure as described above to bootstrap the mean values for conformity during movie-viewing, conformity at rest, the mean correlation between events per minute and conformity (within-participant). Specifically, the input data (e.g., conformity values across participants' movie runs) was resampled with replacement 1000 times. The mean of each sample was recorded to build a sampling distribution of the mean, from which we could obtain the bootstrapped mean value as well as confidence intervals.

**Bootstrapped correlations**. We used a bootstrap approach to determine correlations between transition rate during movie-viewing and at rest, as well as between transition rate, conformity, and neuroticism. Similar to above, this entailed resampling with replacement pairs of values (e.g., movie and rest transition rates) across participants. The correlation between variables for each sample was recorded to build a sampling distribution of correlation, from which we could obtain the bootstrapped correlation value, confidence intervals, and $p$-values. Outliers are identified as values surpassing three median standard deviations and are removed from the analysis[69].

**Conjunction analysis methodology**. To assess regions evoked at transition versus meta-stable timepoints in both task and rest, we first subtracted meta-stable from transition images independently for movie and rest, such that a movie difference image and rest difference image was available for each participant. We masked these images using a gray matter mask, performing comparisons only on those voxels with at least a 50% probability of being gray matter based on an MNI anatomical atlas in the same space[70]. Again, we used a non-parametric boot-strapping analysis, but this time obtained a bootstrap ratio (BSR) image for each of the movie and rest difference images (see e.g., ref. [71]). As the bootstrap ratio approximates a $z$-distribution[72], we used a cumulative distribution function to convert it into a map of voxel-wise $p$-statistics. For each of movie and rest, we thresholded the resulting map at $P < 0.001$, and suppressed supra-threshold voxels within the respective $p$-maps that did not satisfy a minimum cluster extent threshold of 250 voxels (1024 mm$^3$).

To perform conjunction analysis, we thresholded both the movie and rest $p$-maps at $p < 0.05$, setting all voxels above this value to infinity, and computed the product of the $p$-maps. The resulting conjunction $p$-map represented the probability of obtaining a supra-threshold result not only in map A, but also in map B. Because of the initial thresholding step, it had an implied $p$-value threshold of 0.0025. As with the single-task analyses, we then further suppressed supra-threshold voxels within the conjunction $p$-map that did not satisfy a minimum cluster extent threshold of 250 voxels (1024 mm$^3$). We selected these voxel-wise and extent thresholds to achieve a balance between type I and type II error rate[73].

**Statistics and reproducibility**. Most analyses used data from the 7 T Human Connectome Project Young Adult dataset, which involved a large sample and uniquely contained both rest and movie data. Thus, most analyses could not be repeated. Analyses examining the relationship between transition rate and rest and neuroticism were repeated in a second (3 T) data set, as reported in the Results section.

**Reporting summary**. Further information on research design is available in the Nature Research Reporting Summary linked to this article.

## Data availability

Data are available from the Human Connectome Project at humanconnectome.org. Movie-related variables (e.g., event boundaries segmented for this study) are available upon request. A reporting summary for this Article is available as a Supplementary Information File.

## Code availability

Our analysis relied upon algorithms implemented in MATLAB (v.2017a), FreeSurfer (v.5.3), Datavyu (v.1.3.7), FSL (v.5.0.10), R (v.3.6.2), and the Brain Imaging Analysis Kit (RRID: SCR_014824). Custom code used to identify transitions and meta-stable timepoints from the network representation step onwards can be found at https://github.com/j-tseng/neural-transitions.

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

## Acknowledgements

We thank Y. Lu for help with event, sub-event, and cut segmentations; S. Smith for manual hippocampal segmentations; and K. Norman, M. Sabbagh, and G. Blohm for helpful comments. This research was funded by Natural Sciences & Engineering Research Council Discovery Grant 03637 (J.P.), which also supported J.T. Infrastructure funding was provided by the Canada Foundation for Innovation—John R. Evans Leaders Fund (J.P.). and a Queen's University Research Initiation Grant to J.P., who was supported by the Canada Research Chairs program. Data were provided [in part] by the Human Connectome Project, WU-Minn Consortium (Principal Investigators: David Van Essen and Kamil Ugurbil; 1U54MH091657) funded by the 16 NIH Institutes and Centers that support the NIH Blueprint for Neuroscience Research; and by the McDonnell Center for Systems Neuroscience at Washington University.

## Author contributions

Conceptualization, J.T. and J.P.; methodology, J.T. and J.P.; software, J.T. and J.P.; formal analysis, J.T. and J.P.; resources, J.P.; data curation, J.T. and J.P.; writing—original draft, J.T. and J.P.; writing—review and editing, J.T. and J.P.; visualization, J.T. and J.P.; and funding acquisition, J.P.

## Competing interests

The authors declare no competing interests.
