## [Peer Review File · Nature Communications]

Reviewers' comments:

Reviewer #1 (Remarks to the Author):

In this paper the authors propose a new method for detecting transitions in whole-brain patterns of network activity, and apply this method to both movie-watching and resting-state fMRI data. After validating that brain-state transitions are synchronized across subjects during movie-watching, they replicate (and improve upon) earlier work showing that neural transitions correspond to narrative event transitions, and show for the first time that individual transition rate is correlated across movie and resting-state data, and is correlated with trait neuroticism.

I very much enjoyed reading this paper! I think the results are novel and interesting, the use of a large public dataset allows for robust conclusions, and the described method could be a generally-useful approach for studying the dynamics of brain responses during both task and rest. The figures are generally clear, and the quantitative comparison to other possible approaches helps to show empirically that this method outperforms existing approaches on detecting these types of transitions.

Major comments:

1) No code implementing the proposed method was available to review, and the authors state that they "have no new software to report" and do not provide a code repository. Given that one of the primary contributions of this paper is a new (and somewhat complex) analysis method, it is hard to fully evaluate this work without access to the code, and the impact of this publication will be limited if other groups can not easily apply this method.

2) The simulated data is not realistic enough to be meaningful. The simulations (pages 21-23) sample from a normal distribution in t-SNE space, in which every timepoint is independently sampled. This is problematic in two ways: 1) It does not test the properties of the transformation from fMRI data into t-SNE space; 2) It ignores strong temporal autocorrelation present in fMRI signals, which will dramatically change the distribution of step sizes between pairs of points. I would recommend generating more realistic simulated fMRI data without meaningful structure, for example by using phase randomization or `fmrism` from the `brainiak` toolbox.

3) The authors promote as a feature of their method that it does not require hyperparameters that control the number of transitions to identify within each run (e.g. in comparison to other methods on page 30). However, this is not entirely true. The method requires at least three parameters that will change the number of detected transitions or meta-stable points: temporal smoothing with a span of 5 seconds, a peak prominence threshold of 0.06, and a stability width of 10 TRs. Changing these values (especially the peak threshold) will change the number of detected transitions. It would be useful to know how robust their results are to different settings of these parameters.

4) The comparison between the authors' approach and the Hidden Markov Model approach could be improved. On page 14 the authors make three claims about the HMM approach which to my knowledge

are not correct: that it is not grounded in an observable neural phenomenon (I am unsure what this means, but the HMM defines events as patterns of neural activity, similar to the method in this paper); that it cannot be applied to individuals (the HMM can and has been applied to individuals, e.g. individual free recall data); and that it cannot be used in uncharacterized task environments (the HMM could be applied to individual resting-state data as well). In my view the primary distinction between these methods is in scope. The authors' method is designed only to detect whole-brain transition points, and it empirically outperforms the HMM on this task. The HMM can also perform other types of analyses, such as aligning across multiple datasets with the same sequence of states, finding a pre-specified sequence of event patterns in a dataset, and providing probabilistic estimates. Clarifying the strengths of these different approaches would help make the authors' contribution more clear.

Minor comments:

5) It is unclear why the authors refer to stable periods of network activations as "meta-states" that can be "meta-stable." Each meta-state is characterized by a single pattern of activity (not some type of dynamics or collection of patterns) and is interpreted by the authors to correspond to a specific thought, so it's not clear why the "meta-" prefix is necessary.

6) In the discussion of the limitations of previous studies (bottom of page 3 - top of page 4) examples of specific studies should be cited to clarify the approaches that the authors are referring to.

7) On page 11 the authors state that "transition rate was correlated across runs." It is unclear to me what this means - each subject has 4 runs, so is this a correlation across subjects between pairs of runs (and what does the SD indicate)? It seems like a test that explicitly measures across-run versus across-subject variability in some way (e.g. a repeated-measures ANOVA) would be a more direct test of the hypothesis that transition rate is subject-specific.

8) The authors show that participants' head motion was not a large factor in driving transitions for the movie data. Given that motion can be higher during resting-state scans (e.g. Vanderwal et al. (2015). *Inscapes: A movie paradigm to improve compliance in functional magnetic resonance imaging*. NeuroImage) it would be good to confirm that this is also true for the resting-state data.

9) The discussion of the brain regions with transition-related activity (Fig 4) should better connect to the literature on the neural correlates of event transitions. For example, see Speer, N. K., Zacks, J. M., & Reynolds, J. R. (2007). Human brain activity time-locked to narrative event boundaries. *Psychological Science*.

10) The authors state that they are using Mahalanobis distance at multiple points in the paper. This is confusing for two reasons. First is that Mahalanobis distance requires specifying a covariance matrix, and the authors do not state how they are setting this (is it from the empirical covariance across all timepoints?). Second, the t-SNE embedding is defined using a Euclidean metric, such that the distribution of neighbors' Euclidean distances in this space is meant to reflect the distribution of

neighbors' Euclidean distance in the original space. The motivation for switching to a Mahalanobis metric to define step sizes is unclear.

11) The comparison of embedding approaches in Supp Figure 3 is helpful for understanding the critical components of the method. Two other comparisons I'd be interested in seeing: 1) The "unreduced" bar still performs the initial dimensionality reduction to 15 networks. What if no initial dimensionality reduction is applied, i.e. distances are measured in original voxel space? 2) The HMM is applied to a 200-network decomposition, which makes it harder to compare to the authors' proposed method (which uses the 15-network reduction). Does applying the HMM to a 15-network decomposition change the results?

Reviewer #2 (Remarks to the Author):

Tseng and Poppenk present an innovative new method to delineate brain state transitions. This clever method allows the researchers to extract "worms" of thought from continuous neural activity, and thus offers an exciting new technique to quantify the neural dynamics of spontaneous thought. They demonstrate that these brain state transitions are related to event boundaries in movie viewing, and that neuroticism predicts the number of transitions during resting state. The paper is interesting and well-written, and marks a notable advance in the study of spontaneous thought and thought dynamics. I have only a few conceptual and methodological questions/comments that I hope will help to bring out these advances more clearly.

1) I found the connection between neuroticism and transitions interesting, as an attempt to establish a link between the new method and psychology. However, as written, this personality correlation felt like a weak addition to an otherwise methodologically strong paper. Without much background (e.g., in the introduction) on why neuroticism should be related to transitions, the results seem to be justified post hoc, rather than theoretically motivated. Neuroticism is not the first trait that one might expect to correlate with transitions. What about ADHD, or mania, for example? The strength of the correlation is also not particularly strong. The stats are presented as a 1-tailed test, and the correlation would not be significant under a 2-tailed test. Unless this was the only trait measure tested or if this particular hypothesis was preregistered, it would be appropriate to correct for multiple comparisons. From the methods section, it looks like the authors tested the correlation between 5 personality traits and transitions, in which case, the neuroticism correlation would likely not survive correction. If the authors decide to keep this correlation in the paper, it might be helpful to include a graph of the correlation so we can get a sense of what the data look like, perhaps in the supplement.

2) I would like to see some more specific examples of applications of the method in the discussion section. Although the last paragraph of the discussion does mention that it's broadly applicable, I think some examples would drive the point home. What questions could be answered using this technique

that were elusive before? In addition to connecting this work to future research, I'd also like to see the introduction connect to past work on the topic. Readers would benefit from more of a background on the rich history of psychological and philosophical work on the stream of thought, spontaneous thought, mind wandering, thought transitions, etc. (to the extent that space allows).

3) The authors present an interesting finding on the neural regions that differentiate stable time points from transition time points. This finding comes from a conjunction analysis across both movie and rest data. However, to be convinced that transition/stable states are similar across both movie and rest, I would want to see the results from movie and rest separately, rather than in conjunction. The conjunction is indeed interesting, but I think that comparing the movie and rest results side by side would add to our understanding of how these states are instantiated, and would clarify whether or not they are instantiated in similar ways across these two datasets.

4) How can we be sure that transitions reflect changes in thought content – from one thread of thought to a distinct thread of thought – rather than simply changes in cognitive or neural states (e.g., a shift from internal to external attention, or from a high attentional state to a low attentional state)? Could the neural transition measure simply reflect natural variation in shifts from one neural network to another, rather than any particular content shifts? My confusion around this question may stem from the fact that I did not fully understand the analysis of how the features of the movie data related to brain state transitions. Can we take away from these content/event analyses that neural state transitions correspond to changes in content and meaning, rather than a more general transition point? To that end, more explanation of this analysis would be helpful, particularly about the process of extracting feature information from movie, measuring how each feature accounts for the HRF, what comprises the censored vs. uncensored epochs, and how exactly we should interpret these findings, in light of my confusion above.

5) I am curious about the decision to temporally smooth the timeseries with a 5-second moving average filter. Is this a standard procedure when working with data of this type? I am not familiar with this method so it would help to justify or explain the benefits of filtering, and why 5-seconds might be optimal. To what extent are any of the results affected by this filter choice? Most of the analyses will likely be unaffected (e.g., comparisons across movies and resting state). However, the authors present an interesting finding that there are 6.5 transitions per minute. Might we expect to see more thoughts per minute if there weren't a 5-second smoothing filter on the data?

6) One minor issue I found was the claim that meta-state transitions are trait-like because they are consistent within participants across runs (both resting state and movie viewing). While I think that it is indeed compelling and interesting that transitions are consistent within a session, I believe that calling it a trait would require evidence that this is stable over longer periods as well. For example, there is behavioral evidence that thought speed fluctuates with changes in mood. There may be ways to manipulate thought speed within participant. I think the evidence and informativeness of meta state transitions will still be evident without labeling it a trait.

Reviewer #3 (Remarks to the Author):

This is an interesting and fascinating research article that explores the neural correlates of meta-state transitions and stability in the human brain and how these brain time-variant markers relate to individual differences in neuroticism. The fMRI data were obtained while people were scanned 'at rest' or while they were watching some movie clips.

Although I'm generally enthusiastic and supportive of this paper, there are some aspects of the manuscript that require clarification in my opinion. I have also suggested further analyses to improve the quality of an otherwise excellent work.

1) Given the emphasis on individual differences in neuroticism, which is a key personality trait linked to emotional behaviour, I was somehow surprised that the authors did not categorize or tried to divide the 'meaningful' transitions of the event in the movies in terms of their emotional content. How many and which specific transitions in the movie clips were emotional in their content?

2) In the first part of the material & methods section, the authors mention that the full HCP dataset was used as 'training' dataset while the HCP 7T dataset was employed as 'testing' dataset. The terms 'training' and 'testing' dataset are typically used when one aims to discover an effect in a dataset (the training one) and then try to replicate (validate) it in another dataset (testing). This is also known as a train/test split approach. However, this doesn't seem to be the case in this study. I'm thus slightly confused about what the authors are referring to when they describe the 'training' and 'testing' dataset.

3) The effect sizes of neuroticism on meta-state transitions look rather small, ($r=0.15$, -0.18 , one-tailed tests) so I wonder about their statistical reliability. I also suggest the authors to provide a plot of these key findings. Were these results obtained after correction for multiple comparisons?

4) It is also unclear whether other personality traits showed an association with brain measures of transition and stability or whether the results were only specific to neuroticism itself. The authors declare that they have tested the other behavioural traits from the NEO-PI but they only reported the effects of neuroticism. Even if null, the findings for the other personality traits should be included for the sake of clarity and completeness. Furthermore, I wonder whether the statistical model that tested for the associations with personality traits included all the traits simultaneously to account for potential confounding effects driven by the other traits. Or were separate models used for each trait?

5) Were there gender differences in the main findings or was there an interaction between personality traits and gender? There is a rather robust literature showing that neuroticism levels tend to be higher in females than males. This is mirrored by the higher risk of women to develop anxiety- and depression-related disorders. So I wonder whether the neural correlates of meta-state transitions and stability also depend on gender differences and whether there is any personality by gender interactive effect.

6) The authors do not seem to report the brain localization of the effects of neuroticism on the meta-stable brain measures. Did the regions implicated in transitions show even more transitions in people scoring higher in neuroticism? And the reverse for the regions implicated in meta-stability? Can they run a region of interest approach to address this issue?

7) Finally, I suggest the authors to discuss the findings of other papers in the field of personality neuroscience. For example, a recent study (Time-resolved connectome of the five-factor model of personality., Sci Rep. 2019 Oct 21;9(1):15066. doi: 10.1038/s41598-019-51469-2) found that conscientiousness, a trait that is often negatively correlated to neuroticism, is linked to more stable (i.e., reduced time-variant) connectivity patterns across a series of resting-state networks. This brings back the question of whether other traits over and above neuroticism are linked to meta-stable brain measures. Even if this is not the case, the above study could still form the basis of a working hypothesis for the current study which is at the moment not clearly specified.

Reviewer #1

In this paper the authors propose a new method for detecting transitions in whole-brain patterns of network activity, and apply this method to both movie-watching and resting-state fMRI data. After validating that brain-state transitions are synchronized across subjects during movie-watching, they replicate (and improve upon) earlier work showing that neural transitions correspond to narrative event transitions, and show for the first time that individual transition rate is correlated across movie and resting-state data, and is correlated with trait neuroticism.

I very much enjoyed reading this paper! I think the results are novel and interesting, the use of a large public dataset allows for robust conclusions, and the described method could be a generally-useful approach for studying the dynamics of brain responses during both task and rest. The figures are generally clear, and the quantitative comparison to other possible approaches helps to show empirically that this method outperforms existing approaches on detecting these types of transitions.

We thank the reviewer for their constructive feedback and appreciate the enthusiastic response. We, too, are optimistic that the paper will be broadly useful and applicable within the domain of cognitive neuroscience. We attempt to address their concerns in our comments below.

Reviewer #1, Comment 1

No code implementing the proposed method was available to review, and the authors state that they "have no new software to report" and do not provide a code repository. Given that one of the primary contributions of this paper is a new (and somewhat complex) analysis method, it is hard to fully evaluate this work without access to the code, and the impact of this publication will be limited if other groups can not easily apply this method.

We have now uploaded the MATLAB code to a Github repository as per the reviewer's suggestion.

Manuscript Changes: We have amended the Code Availability statement (p. 41) to reflect availability of the analysis code.

Reviewer #1, Comment 2

The simulated data is not realistic enough to be meaningful. The simulations (pages 21-23) sample from a normal distribution in t-SNE space, in which every timepoint is independently sampled. This is problematic in two ways: it does not test the properties of the transformation from fMRI data into t-SNE space; it ignores strong temporal autocorrelation present in fMRI signals, which will dramatically change the distribution of step sizes between pairs of points. I would recommend generating more

realistic simulated fMRI data without meaningful structure, for example by using phase randomization or fmrisim from the brainiak toolbox.

We appreciate the reviewer’s suggestion on using realistic simulated fMRI data. As suggested, we used fmrisim from the BrainIAK toolbox to generate a noise dataset consisting of phase randomized participant data for the first movie run. Phase shifts were carried out voxel-wise for each participant (i.e., random different phase shifts instead of the same across all voxels). After transforming the fMRI signal into the network representation through dual regression of the 15-network spatial ICA maps, we followed the method outlined in our paper (i.e., ran 100 iterations of the t-SNE algorithm, calculated Mahalanobis distances). The final step distance vector for each “noise” participant consisted of the mean step distance vector across these 100 iterations.

Inspecting the individual t-SNE iterations on the noise data, we notice that results often consist of a few lengthy contiguous clusters (i.e., few smaller transitions), in contrast to the several contiguous clusters we see in the t-SNE projections of real data that result in numerous identified transitions. Furthermore, comparing a participant’s mean noise step distance vector against their real mean step distance vector, we notice that the real data yields transitions that are more consistent across repeated t-SNE algorithm runs. These results highlight the importance of carrying out the repeated t-SNE algorithms step to stabilize the transitions that are subsequently identified.

We also used this noise dataset to conduct a secondary analysis investigating the association between noise transitions and movie features. These results provide a baseline comparison to the real eta-squared values we obtained which describe the alignment of transitions to features. We found no significant association between transitions and movie features.

Supplementary Fig. 2. Comparison of noise and real mean step distance vector for a participant in the first movie run. A noise dataset was generated by phase randomizing in a voxel-wise fashion each participant’s Movie 1 fMRI run. The same procedure (dual regression, 100 iterations of the t-SNE algorithm, Mahalanobis distance calculation) was carried out to obtain mean noise step distance vectors. As seen above, t-SNE space representations

of noise data do not jump at consistent timepoints across repeated iterations, resulting in smaller transition amplitudes than in the real step distance vector.

Manuscript Changes: We have now included discussion on the results described above instead of the original noise analysis (p. 22-23, Supplementary Fig. 2), as well as integrated the movie feature analysis results in the relevant section of the manuscript (p. 9, 26).

Reviewer #1, Comment 3

The authors promote as a feature of their method that it does not require hyperparameters that control the number of transitions to identify within each run (e.g. in comparison to other methods on page 30). However, this is not entirely true. The method requires at least three parameters that will change the number of detected transitions or meta-stable points: temporal smoothing with a span of 5 seconds, a peak prominence threshold of 0.06, and a stability width of 10 TRs. Changing these values (especially the peak threshold) will change the number of detected transitions. It would be useful to know how robust their results are to different settings of these parameters.

To evaluate the importance of these parameter settings, we tested the effects of varying minimum peak prominence (MPP) values and temporal smoothing spans on the movie feature analysis.

Minimum Peak Prominence: We derived sets of transition and meta-stable timepoints from 11 MPP values: [0.01, 0.02, ..., 0.11]. These timepoints were fed into the movie feature analysis. Results are shown in Supplementary Fig. 3a below, with results of the original, moderate MPP value (0.06) represented by the middle bar in each 11-bar cluster. As we aimed to identify neural transitions that corresponded with a new thought, which we proposed are analogous to event boundaries and semantic content changes in movies, we are reassured that the alignment between transitions and events remained stable across MPP values. Furthermore, in the semantic feature, alignment strength increases as values approach the actual MPP value (0.06), but then remain stable for values greater than the actual MPP value.

Temporal Smoothing Span: Smoothing occurs on the 15-network activity configuration, just before this representation is used as input into the dimensionality reduction algorithm. We originally used a five-second span, since this is close to the time-to-peak in the hemodynamic response function. However, to probe this parameter, we derived meta-state space representations for data smoothed with the following spans: [1 s, 3 s, 5 s, 7 s, 9 s]. From these five sets of data, we derived five sets of transition and meta-stable timepoints. As above, these timepoints were fed into the movie feature analysis. Results are shown in Supplementary Fig. 3b below, with results of the original span (span = 5 s) represented by the middle bar in each 5-bar cluster.

Supplementary Fig. 3. Effects of varying minimum peak prominence threshold and smoothing spans. Eta-squared values describing the proportion of variance in movie features explained by alignment to transition vs. meta-stable timepoints. Note that all auditory feature results showed stronger association to meta-stable rather than transition timepoints. **(a)** Comparison of eta-squared results for varying minimum peak prominence. **(b)** Comparison of eta-squared results for varying temporal smoothing spans.

Stability Width: We note that the purpose of setting the stability width was to gather a sample of stable timepoints of appropriate quantity for comparison against transition timepoints, rather than to pinpoint *every* period of meta-stability. By making this window as long as possible while still retaining a sufficient quantity of meta-stable timepoints, we sought to optimize our ability to cleanly characterize those timepoints

while retaining sufficient statistical power to obtain a robust parameter estimate for each participant. To this end, we selected a stability width of 10 s, which yielded approximately 40 meta-stable timepoints (see Response Fig. 1 for a histogram describing the number of meta-stable periods identified across participants and runs).

Response Fig. 1. Distribution of the number of meta-stable timepoints identified using a stability width of 10 s.

Manuscript Changes: Results from testing different minimum peak prominence thresholds and smoothing spans are now included in the manuscript on pages 33-34.

Reviewer #1, Comment 4

The comparison between the authors' approach and the Hidden Markov Model approach could be improved. On page 14 the authors make three claims about the HMM approach which to my knowledge are not correct:

- a. that it is not grounded in an observable neural phenomenon (I am unsure what this means, but the HMM defines events as patterns of neural activity, similar to the method in this paper);*
- b. that it cannot be applied to individuals (the HMM can and has been applied to individuals, e.g. individual free recall data);*
- c. and that it cannot be used in uncharacterized task environments (the HMM could be applied to individual resting-state data as well).*

In my view the primary distinction between these methods is in scope. The authors' method is designed only to detect whole-brain transition points, and it empirically outperforms the HMM on this task. The HMM can also perform other types of analyses, such as aligning across multiple datasets with the same sequence of

states, finding a pre-specified sequence of event patterns in a dataset, and providing probabilistic estimates. Clarifying the strengths of these different approaches would help make the authors' contribution more clear.

We agree this point requires careful clarification and are glad the reviewer has raised it. The key distinction we wished to convey is with regards to the involvement of priors: HMM requires them, whereas the current approach only involves priors only at the most preliminary stage for generating our preferred structure of input signals (using network masks that can be adapted from HCP-1200 in order to reduce the dimensionality of fMRI timeseries).

For example, where it has been applied to individuals, HMM was first fitted to either group data (to learn template states, or at least number of states to be potentially detected in an individual's run), or fitted to an earlier run of the same participant (to learn template states to be potentially detected in free recall). By contrast, our approach can be applied to the very first pilot participant's very first fMRI run. Where it has been used in uncharacterized task environments, HMM was trained to recognize very broad states (e.g., Chen et al., 2016 detected 9 states); our approach requires no a priori assumptions about which states will be visited.

This can be regarded as a strength or weakness of either approach, depending on how the technique is to be used. Where quality priors can be leveraged, HMM is able to use them to narrow focus to specific relevant states. This is advantageous, for example, in the free recall experiment described by the reviewer, where specific, pre-defined states are of interest. Where it is harder to predict which states will be visited, such as in resting-state data, the current technique is advantageous. For this reason, we have long hoped to downplay the "horse-race" interpretation invited by Supplementary Fig. 4: regardless of how algorithms perform relative to one another with respect to alignment to movie features, HMM and the current approach are ultimately qualitatively different approaches with different applications (even in the current paper).

Our point about grounding in a neural phenomenon refers to our observation of "transition" regions in our conjunction analysis, whereas we're not aware of evidence describing such a distinction using HMM. This is auxiliary to our main point above, however, so we have removed this comment.

We think we are ultimately agreeing with the reviewer's conclusion that the principle difference between the approaches is in scope, but felt it important to be clear about how we see the relationship between the approaches.

Manuscript Changes: We have made the above clarifications in the discussion section (p. 16-17).

Reviewer #1, Comment 5

It is unclear why the authors refer to stable periods of network activations as "meta-states" that can be "meta-stable." Each meta-state is characterized by a single pattern of activity (not some type of dynamics or collection of patterns) and is interpreted by the authors to correspond to a specific thought, so it's not clear why the "meta-" prefix is necessary.

This terminology is frequently used in the state analysis literature: for example, Miller and colleagues (2016) refer to their lower-dimensional representation of brain activity as a meta-state space because it is a higher order representation. Here, similarly, each two-dimensional timepoint corresponds to patterns of activation across 15 networks. We also prefer the use of this term because, in some sciences, *stable* characterizes a global minimum with respect to an energy surface, whereas *meta-stable* reflects a local minimum. The latter better characterizes the nature of states in the brain, where there is no true "global minimum", but rather a set of states that may reflect a local minimum on the energy surface at some point in time.

Manuscript Changes: We have now added these observations to the main text (p. 4), as well as further explanation in the Methods section (p. 20).

Reviewer #1, Comment 6

In the discussion of the limitations of previous studies (bottom of page 3 - top of page 4) examples of specific studies should be cited to clarify the approaches that the authors are referring to.

We regret this oversight and have now added these references to clarify the approaches that we were referring to (p. 4).

Reviewer #1, Comment 7

On page 11 the authors state that "transition rate was correlated across runs." It is unclear to me what this means - each subject has 4 runs, so is this a correlation across subjects between pairs of runs (and what does the SD indicate)? It seems like a test that explicitly measures across-run versus across-subject variability in some way (e.g. a repeated-measures ANOVA) would be a more direct test of the hypothesis that transition rate is subject-specific.

One goal of our analysis was to evaluate the stability of our measures across the four rs-fMRI runs. To do this, we took the transition rate measurement for each subject in rs-fMRI run 1, then computed the correlation of this vector against the analogous one obtained for rs-fMRI run 2. We did this for all six possible pairings of runs. What we originally reported was the average and standard deviation across the resulting six

pairwise correlation coefficients. We acknowledge, however, that this was a rather crude approach and that other statistics would be better suited to this analysis.

To this end, we now adopt an interrater reliability approach. Intraclass correlation analysis is designed to evaluate the degree of agreement in “rater scores” (or in this case, transition rate as observed in different runs) relative to the degree of variation within rater (or in this case, transition rate across subjects). This test statistic is therefore ideally suited for evaluating relationships in question in the manner proposed by the reviewer. Further, it does not require the awkward step of taking a mean and standard deviation, since all runs are evaluated simultaneously.

Accordingly, we fed the transition rate measurements for 184 participants across all 8 runs (4 resting state and 4 movie-viewing) into an ICC function implemented in the *psych* package in R. As the variance across days stemming from, for example, mood, could be interpreted as a random rater effect, we chose the two-way random, average score of consistency. We found transition rate to be highly reliable across the eight runs, $ICC(2,k) = 0.86$, 95% CI: [0.84, 0.89], $F(183,1288) = 7.6$, $p < .001$.

Manuscript Changes: We have now provided used ICC to quantify the consistency of transition rate across rest runs, as well as provided other quantifications of consistency (p. 11).

Reviewer #1, Comment 8

The authors show that participants' head motion was not a large factor in driving transitions for the movie data. Given that motion can be higher during resting-state scans (e.g. Vanderwal et al. (2015). Inscapes: A movie paradigm to improve compliance in functional magnetic resonance imaging. NeuroImage) it would be good to confirm that this is also true for the resting-state data.

We imported the transitions and meta-stable timepoints identified at rest and relative head motion from the rest runs. Next, we used the same strategy as with the movie feature analysis. In particular, we reasoned that if participants' head motion causes transitions, then the average value of relative head motion at a transition timepoint should be significantly larger than the average value at a meta-stable timepoint. We obtained the value representing this transition-meta-stable head motion difference for each participant, then fed these values into a bootstrap analysis comparing against the null hypothesis of zero. We found no significant association between transitions and head motion at rest, mean transition-meta-stable step distance difference = -0.00, BSR = -1.10, $p = .273$, 95% CI: [-0.00, 0.00].

Manuscript Changes: We now report these findings in the supplemental section (p. 36).

Reviewer #1, Comment 9

The discussion of the brain regions with transition-related activity (Fig 4) should better connect to the literature on the neural correlates of event transitions. For example, see Speer, N. K., Zacks, J. M., & Reynolds, J. R. (2007). Human brain activity time-locked to narrative event boundaries. Psychological Science.

As requested, we have now included this reference along with more discussion on the overlap in neural bases of event cognition and spontaneous thought in the introduction (p. 5).

Reviewer #1, Comment 10

The authors state that they are using Mahalanobis distance at multiple points in the paper. This is confusing for two reasons. First is that Mahalanobis distance requires specifying a covariance matrix, and the authors do not state how they are setting this (is it from the empirical covariance across all timepoints?). Second, the t-SNE embedding is defined using a Euclidean metric, such that the distribution of neighbors' Euclidean distances in this space is meant to reflect the distribution of neighbors' Euclidean distance in the original space. The motivation for switching to a Mahalanobis metric to define step sizes is unclear.

We noticed from visual inspection that the reduced 2D representations sometimes had diagonal temporal drift, suggesting some covariance between the two axes. When using Mahalanobis, but not Euclidean distance, any covariance between dimensions is accounted for. Thus, we decided to use the Mahalanobis distance metric to derive the step distance vectors. Here, covariance matrices are empirical (i.e., calculated directly from the inputted data).

As the decision to use the Mahalanobis distance metric was based on the temporal drift observation, we did not consider changing the internal t-SNE distance metric. To compare the two options, we re-ran the dimensionality reduction algorithm with the Mahalanobis distance option and proceeded with the same steps to obtain a set of transition and meta-stable timepoints from the participants' movie-viewing data. Then, we fed these timepoints into the movie feature analysis. Results pictured in Response Fig. 2 show minimal differences in the resulting alignment to movie features between the two options.

Manuscript Changes: We now clarify that the covariance matrices are empirical (i.e., based on the sample) on page 21.

Response Fig. 2. Comparison of movie feature analysis for transition and meta-stable timepoints derived from implementing Euclidean distance (gray) or Mahalanobis distance (purple) within the t-SNE algorithm.

Reviewer #1, Comment 11

The comparison of embedding approaches in Supp Figure 3 is helpful for understanding the critical components of the method. Two other comparisons I'd be interested in seeing: 1) The "unreduced" bar still performs the initial dimensionality reduction to 15 networks. What if no initial dimensionality reduction is applied, i.e. distances are measured in original voxel space? 2) The HMM is applied to a 200-network decomposition, which makes it harder to compare to the authors' proposed method (which uses the 15-network reduction). Does applying the HMM to a 15-network decomposition change the results?

1. Original Voxel Space: Although we attempted to implement the proposed analysis, we found that it was not possible to apply the Mahalanobis distance metric on a voxel-wise basis due to technical constraints. As the distance metric uses a sample covariance matrix, it attempts to calculate it for 298,307 voxels (i.e., 298,307 by 298,307), greatly exceeding computational limits. Furthermore, to our knowledge, few methods use the voxel-wise neural data directly as input as individual noisy voxels are not sufficiently down-regulated to limit their impact on the analysis. Therefore, although it is an interesting idea, we regret this specific suggestion is not tractable.

2. HMM Applied to 15-Network Space: We originally applied HMM to the 200-network reduction because this was most analogous to past use (Baldassano et al.,

2016). However, as suggested, we also tried applying HMM to the 15-network reduction (HMM-15) instead of the 200-network reduction (HMM-200), and carried the transition and meta-stable periods forward into the movie feature analysis. This was a great suggestion, as transitions identified for HMM-15 were more strongly associated to movie features than HMM-200, suggesting that the 15-network formulation may be an advantageous representation of the data for multiple forms of analysis. However, our t-SNE method still consistently outperformed both HMM analyses (see Supplementary Fig. 4 enclosed below).

We have included the Hidden Markov Model approach results that use the 15-network decomposition in Supplementary Fig. 4, as well as modified discussion surrounding those results in the Methods section (p. 27-28, 30-31).

Supplementary Fig. 4. Impact of embedding approach on transitions' predictive power for movie features. Eta-squared values describing the proportion of variance in uncensored movie features explained by alignment to transition vs. meta-stable timepoints. Filled bars denote features that are aligned to transitions, whereas empty bars denote stronger alignment to meta-stability. Colours indicate the method used to identify transition and meta-stable timepoints.

Reviewer #2

Tseng and Poppenk present an innovative new method to delineate brain state transitions. This clever method allows the researchers to extract “worms” of thought from continuous neural activity, and thus offers an exciting new technique to quantify the neural dynamics of spontaneous thought. They demonstrate that these brain state transitions are related to event boundaries in movie viewing, and that neuroticism predicts the number of transitions during resting state. The paper is interesting and well-written, and marks a notable advance in the study of spontaneous thought and thought dynamics. I have only a few conceptual and methodological questions/comments that I hope will help to bring out these advances more clearly.

We are happy that the reviewer found the paper to be interesting and well-written. We have addressed the methodological and conceptual questions raised below, which we believe have improved the paper.

Reviewer #2, Comment 1

I found the connection between neuroticism and transitions interesting, as an attempt to establish a link between the new method and psychology. However, as written, this personality correlation felt like a weak addition to an otherwise methodologically strong paper. Without much background (e.g., in the introduction) on why neuroticism should be related to transitions, the results seem to be justified post hoc, rather than theoretically motivated. Neuroticism is not the first trait that one might expect to correlate with transitions. What about ADHD, or mania, for example? The strength of the correlation is also not particularly strong. The stats are presented as a 1-tailed test, and the correlation would not be significant under a 2-tailed test. Unless this was the only trait measure tested or if this particular hypothesis was preregistered, it would be appropriate to correct for multiple comparisons. From the methods section, it looks like the authors tested the correlation between 5 personality traits and transitions, in which case, the neuroticism correlation would likely not survive correction. If the authors decide to keep this correlation in the paper, it might be helpful to include a graph of the correlation so we can get a sense of what the data look like, perhaps in the supplement.

As suggested by the reviewer, we have included more background in the paper regarding the neuroticism hypothesis and emphasize the theories it draws on in personality research. We also clarify our explicit hypotheses for correlations between neuroticism and the measures of mental dynamics (transition rate and conformity) in the Methods section. We considered other personality correlations to be exploratory and report them in our supplemental text for descriptive purposes, as requested by R3. We now also supply a plot (Supplementary Fig. 5a).

We recognize the result may invite skepticism, given the small size of the correlation and ready availability of multiple personality dimensions. Accordingly, we took further steps to confirm the robustness of our result. In particular, it occurred to us that it would be appropriate to attempt to replicate our finding by using rs-fMRI and personality data associated with the (much larger) 3T dataset, which included 4 resting state runs per person. We followed the same procedures as in our initial analysis to obtain a transition rate for each participant in this new dataset. Bootstrap correlations revealed that average transition rate was again modestly, but robustly, correlated with neuroticism, $r(970) = 0.09$, $p = .006$, 95% CI: [0.02, 0.15], thereby confirming our initial findings in a powerful second sample.

In addition, we note that results were less marginal with respect to the relationship between neuroticism and our other key dependent variable, conformity, further solidifying the relevance of neuroticism to our transition metrics. As with transition rate, we hypothesized that a noisier mind would result in lower conformity, as the individual would have more idiosyncratic, internally triggered transitions compared to the group. The 3T dataset did not contain the necessary mv-fMRI data to replicate this 7T finding, but the result was also associated with a higher level of statistical significance, $p = 0.008$ (which would survive multiple corrections comparison even if one were skeptical about our hypotheses).

These results are now integrated in-text on page 13.

Graph of the Correlation

Supplementary Fig. 5a. Correlation between transition rate and neuroticism in 7T dataset. Transition rate is calculated as the total number of transitions divided by the time in minutes. Neuroticism scores are obtained from the NEO Five Factor Inventory administered to HCP participants.

On Examining ADHD, Mania, or Other Thought Disorders: On the strength of the above findings, and the convergence of neuroticism with our amygdala results, we opted to retain our focus on neuroticism for the current paper. This allowed us to keep our analyses focused on the Human Connectome Project dataset, which is distinctive for featuring both a large number of participants as well as both movie and resting data

(a crucial feature for our analysis), but that is limited to healthy adults (Essen et al., 2013). Thus, it is unfortunately not possible for us to examine correlations between transitions and individuals with more obvious (but subsequently excluded) mental disorders, although this is clearly an excellent suggestion for future analysis (which we have included in our discussion section (p. 17-18)).

Reviewer #2, Comment 2

I would like to see some more specific examples of applications of the method in the discussion section. Although the last paragraph of the discussion does mention that it's broadly applicable, I think some examples would drive the point home. What questions could be answered using this technique that were elusive before? In addition to connecting this work to future research, I'd also like to see the introduction connect to past work on the topic. Readers would benefit from more of a background on the rich history of psychological and philosophical work on the stream of thought, spontaneous thought, mind wandering, thought transitions, etc. (to the extent that space allows).

As suggested, we have now included more background in the introduction (p. 3-4), and now describe additional unique applications to the discussion (p. 18):

By lending a level of validity and reliability to measuring thought dynamics that were unavailable using past introspective approaches, our approach also creates opportunities to understand cognition. Although our analysis has focused on movie and resting fMRI data from healthy adults, our methods are applicable to a wide range of tasks and populations, as well as even case studies, as no group data are required. To illustrate some unique affordances, one can ask other individual-difference questions such as: does transition rate influence a person's ability to remain engaged in sustained attention task? Or, using a more task-based approach: while it is known that novel stimuli are initially attention-grabbing, are there differences in the thought dynamics associated with watching a favourite movie for the first time, relative to the fifth time? Regarding special populations, can measures of thought dynamics serve a clinical function by offering early detection of disordered thought in schizophrenia, or rapid thought in ADHD or mania?

Reviewer #2, Comment 3

The authors present an interesting finding on the neural regions that differentiate stable time points from transition time points. This finding comes from a conjunction analysis across both movie and rest data. However, to be convinced that transition/stable states are similar across both movie and rest, I would want to see the results from movie and rest separately, rather than in conjunction. The conjunction is indeed interesting, but I think that comparing the movie and rest results side by side

would add to our understanding of how these states are instantiated and would clarify whether or not they are instantiated in similar ways across these two datasets.

We have now added separate whole-brain analyses for rest, movie, in addition to the original conjunction analysis. A comprehensive description of the voxel clusters can be found in Supplementary Tables 1, 2 and 3 of the manuscript. Inspection of the brain regions activated at transitions during movie-viewing vs. at rest show consistency across task contexts, as is formalized in the conjunction analysis (Fig. 4 of the main manuscript, also enclosed below). Briefly, an analogous pattern of midline activations in the anterior cingulate cortex (ACC), posterior cingulate cortex (PCC) and the cuneus are clearly observed across the two analyses. The insula, precuneus and supramarginal gyrus were also powerfully and consistently implicated. By contrast, meta-stable activations were consistently observed in dorsal and lateral frontal and parietal regions, along with one dorsomedial region.

Fig. 4. Spontaneous thought and attention regions distinguish transition from meta-stable timepoints in both task and rest. (a) During movie-viewing fMRI, greater BOLD activity was observed during transitions than meta-stable timepoints, in several midline regions, including anterior cingulate cortex, posterior cingulate cortex, and visual association cortex. (b) During resting-state fMRI, greater BOLD activity was observed during transitions than meta-stable timepoints in similar midline regions. (c) Substantiating the subjective similarity of (a) and (b), conjunction analysis revealed substantial voxel-wise overlap for regions activated during transitions (see Supplementary Tables 1-3 for voxel cluster details).

One may notice that the left and right insula are associated with transitions in the conjunction analysis, and yet one hemisphere or the other is absent from each task. This can be explained by the fact that spatial extent thresholding was run separately for each contrast. To avoid confusion around this issue, in Supplementary Table 3, we report activation for each individual task at the peak coordinate of each conjunction cluster. By definition, each conjunction cluster involved activation in both tasks.

Further to the figure and tables described above, these new analyses are now reported in the paper (p. 11).

Reviewer #2, Comment 4

How can we be sure that transitions reflect changes in thought content – from one thread of thought to a distinct thread of thought – rather than simply changes in cognitive or neural states (e.g., a shift from internal to external attention, or from a high attentional state to a low attentional state)? Could the neural transition measure simply reflect natural variation in shifts from one neural network to another, rather than any particular content shifts? My confusion around this question may stem from the fact that I did not fully understand the analysis of how the features of the movie data related to brain state transitions. Can we take away from these content/event analyses that neural state transitions correspond to changes in content and meaning, rather than a more general transition point? To that end, more explanation of this analysis would be helpful, particularly about the process of extracting feature information from movie, measuring how each feature accounts for the HRF, what comprises the censored vs. uncensored epochs, and how exactly we should interpret these findings, in light of my confusion above.

To address the reviewers' questions, as the reviewer intuitively, our linking of the psychological meaning of the network transitions identified in our analysis rests on the alignment of the transitions to movie features. Based on the observation that the transitions have various similar properties at rest relative to the movie analysis, we then argue that it is reasonable to generalize the psychological properties as well, an idea we further support by demonstrating psychological of transitions in the resting data through our neuroticism analysis.

To better scaffold this logic for readers, we have made the following changes:

- We now define spontaneous thought, thought, and mental state in the Introduction section (p. 3).
- We now discuss in the introduction the parallels between spontaneous thought and event segmentation, to provide reasoning for why alignment of network transitions and event boundaries provide initial evidence that transitions reflect thought transitions at rest (p. 5).

- We have added further details to the main manuscript on the feature analysis procedure (p. 9), including discussion of our censoring analysis.
- We have added a reference directing readers to the *Movie feature vectors* section in the Methods for a full description of how feature information was extracted (including information about the HRF, etc.; see p. 9 and 26.).
- Added reference in main text pointing readers to the *Disentangling event boundaries and movie features* section in the Methods for an outline of the feature vector censoring process (p. 9).

Reviewer #2, Comment 5

I am curious about the decision to temporally smooth the timeseries with a 5-second moving average filter. Is this a standard procedure when working with data of this type? I am not familiar with this method so it would help to justify or explain the benefits of filtering, and why 5-seconds might be optimal. To what extent are any of the results affected by this filter choice? Most of the analyses will likely be unaffected (e.g., comparisons across movies and resting state). However, the authors present an interesting finding that there are 6.5 transitions per minute. Might we expect to see more thoughts per minute if there weren't a 5-second smoothing filter on the data?

There is no standard smoothing procedure when working with the network representation of brain activity, so we chose to smooth as a first-pass method of filtering out small fluctuations in network activity that could interfere with the dimensionality reduction. We selected a span of 5 seconds as it approximated the length of the canonical hemodynamic response function and was also long enough to effectively dampen small fluctuations.

To test the effects of filter choice, we derived meta-state space representations for data smoothed with the following spans: [1 s, 3 s, 5 s, 7 s, 9 s]. From these five sets of data, we derived five sets of transition and meta-stable timepoints. As above, these timepoints were fed into the movie feature analysis. Results are shown in the figure below, with results of the original span (span = 5 s) represented by the middle bar in each 5-bar cluster revealing that this chosen span yields transitions that align most strongly with movie features.

Supplementary Fig. 3b. Effects of varying smoothing spans. Comparison of eta-squared results for varying temporal smoothing spans.

We also determined transition rate for each of these five spans. A one-way ANOVA revealed that there was a significant difference in mean transition rate between the spans. Our chosen span was also often (but not always) optimal from a signal detection perspective. However, follow-up pairwise t-tests between our chosen span (5 seconds) and each of the other spans revealed no significant differences (all $ps > 0.27$).

In conclusion, an exploration of the parameter space confirmed that our selection of this smoothing window was desirable in terms of the alignment of our measures to features, yet did not elicit a reliable difference on the analysis outcome relative to no temporal smoothing (i.e., span = 1 s). We have updated the supplemental methods to characterize this information about parameter choice (p. 23-24).

Reviewer #2, Comment 6

One minor issue I found was the claim that meta-state transitions are trait-like because they are consistent within participants across runs (both resting state and movie viewing). While I think that it is indeed compelling and interesting that transitions are consistent within a session, I believe that calling it a trait would require evidence that this is stable over longer periods as well. For example, there is behavioral evidence that thought speed fluctuates with changes in mood. There may be ways to manipulate

thought speed within participant. I think the evidence and informativeness of meta state transitions will still be evident without labeling it a trait.

It is important to note that the sessions in our analysis fell both within- and across-days (Day 1: [Rest 1, Rest 2, Movie 1, Movie 2], Day 2: [Rest 3, Rest 4, Movie 3, Movie 4]). We have now added clarification on the breakdown in-text to emphasize that the reported correlation is across a longer period than a single day (p. 12).

Furthermore, we now explicitly report agreement of our measures within and across days. First, we calculated the stability of transition rate *within* each day (i.e., pairwise correlations between runs that took place on the same day), and found that transition rate was correlated within sessions (Day 1: $ICC(2,k) = 0.79$, 95% CI: [0.74, 0.83], $F(183,549) = 4.6$, $p < .001$; Day 2: $ICC(2,k) = 0.80$, 95% CI: [0.75, 0.84], $F(183,549) = 5.2$, $p < .001$). Next, we calculated the average transition rate on each Day 1 and Day 2, and found that transition rate was correlated across days, $r = 0.64$, $p < .001$, 95% CI: [0.54, 0.73]. Finally, we examined overall reliability across all 8 runs. We again found transition rate to be highly consistent, $ICC(2,k) = 0.86$, 95% CI: [0.84, 0.89], $F(183,1288) = 7.6$, $p < .001$. Overall, these results strengthen the case that our measures are trait-like: mood and other state effects can be expected to vary from one day to the next, whereas we found transition rate to be consistent across days. These results are now integrated into the text (p. 12-13).

Reviewer #3

This is an interesting and fascinating research article that explores the neural correlates of meta-state transitions and stability in the human brain and how these brain time-variant markers relate to individual differences in neuroticism. The fMRI data were obtained while people were scanned 'at rest' or while they were watching some movie clips.

Although I'm generally enthusiastic and supportive of this paper, there are some aspects of the manuscript that require clarification in my opinion. I have also suggested further analyses to improve the quality of an otherwise excellent work.

We are happy the reviewer was so enthusiastic, and address each of their suggested analyses and requested clarifications below.

Reviewer #3, Comment 1

Given the emphasis on individual differences in neuroticism, which is a key personality trait linked to emotional behaviour, I was somehow surprised that the authors did not categorize or tried to divide the 'meaningful' transitions of the event in the movies in terms of their emotional content. How many and which specific transitions in the movie clips were emotional in their content?

We are grateful for the reviewer's suggestion that emotion is a valuable feature to link to our metrics. Based on this suggestion, we performed several new analyses. Before describing these, it is important to note that "emotion" does not fit in well with the table of movie features (describing the relation of transitions to events, actions, etc.) because it is rarely possible to pin emotion to a specific time point. Instead, we evaluated this relationship by evaluating the relationship between our metrics and the emotionality of each clip as a whole.

With respect to our specific predictions, perception is biased towards arousing stimuli (Mather & Sutherland, 2011). An emotionally arousing film stimulus should, therefore, increase the amount of control exerted by the stimulus over participants' perceptions and, in so doing, decrease the rate of spontaneous, stimulus-independent cognitions. We operationalized this hypothesis as follows: clips with higher arousal ratings will be associated with lower transition rates and greater alignment of transitions across participants.

To establish arousal ratings for the film clips, we gathered responses from 12 first year psychology students who watched, in a random sequence, the 14 movie clips that were presented during the mv-fMRI runs. After viewing each film clip, participants evaluated the arousal of the clip. We used these ratings to establish an average arousal rating for each film clip. This resulted in a vector of 14 average arousal scores.

Returning to the brain data, we constructed a parallel 14-clip transition rate vector for each mv-fMRI participant. We then calculated the correlation between the arousal vector and each participant's transition rate vector. This resulted in a correlation coefficient for each participant that described the relationship between the level of arousal associated with each clip, and the participants' resulting transition rate. Inputting this distribution of coefficients into a bootstrap analysis, we observed that higher arousal was reliably associated with fewer transitions, mean $r = -0.16$, BSR = -8.83, $p < .001$.

We also obtained a 14-clip conformity vector describing the degree of similarity between the participant's step distance vector and the group for each clip. We calculated the correlation between the arousal vector and each participant's conformity vector. The distribution of correlation values describing the relationship between arousal and conformity was fed into a bootstrap analysis. We found that higher arousal was associated with more conformity, mean $r = 0.10$, BSR = 4.73, $p < .001$.

Taken together, our analysis confirmed our hypothesis that more emotional stimuli would better capture and maintain the viewer's attention, resulting in fewer overall transitions (i.e., lower transition rate) and higher similarity to the group (i.e., conformity). Intriguingly, this interpretation is also consistent with our observation that the amygdala is associated with meta-stability during movie-viewing.

Results from this analysis are now integrated into the supplemental section of the manuscript (p. 37-38).

Reviewer #3, Comment 2

In the first part of the material & methods section, the authors mention that the full HCP dataset was used as 'training' dataset while the HCP 7T dataset was employed as 'testing' dataset. The terms 'training' and 'testing' dataset are typically used when one aims to discover an effect in a dataset (the training one) and then try to replicate (validate) it in another dataset (testing). This is also known as a train/test split approach. However, this doesn't seem to be the case in this study. I'm thus slightly confused about what the authors are referring to when they describe the 'training' and 'testing' dataset.

We regret the confusion surrounding our use of these terms. We referred to the 3T dataset as the training dataset because the spatial ICA maps that allow us to transform fMRI signal into the network representations were derived from it. However, the reviewer is correct that there is nothing about our analysis that involves a train/test dynamic. Therefore, we now simply refer to 3T and 7T datasets.

Reviewer #3, Comment 3

The effect sizes of neuroticism on meta-state transitions look rather small, ($r=0.15$, -0.18 , one-tailed tests) so I wonder about their statistical reliability. I also

suggest the authors to provide a plot of these key findings. Were these results obtained after correction for multiple comparisons?

Before addressing this comment, it is important to note that our hypotheses specifically targeted neuroticism, such that we did not correct for multiple comparisons. This is now clarified in our Results and Discussion sections, which provide more background on the neuroticism hypothesis and emphasize that it drew on theories in personality research. We also clarify our explicit hypotheses for correlations between neuroticism and the measures of mental dynamics (transition rate and conformity) in the Methods section. We considered our other personality correlations to be exploratory and included them in our supplemental text for descriptive purposes. As requested, we now also supply a plot (Supplementary Fig. 5a).

These points made, we recognize the result may invite skepticism, given the small size of the correlation and ready availability of multiple personality dimensions. Accordingly, we took further steps to confirm the robustness of our result. In particular, it occurred to us that it would be appropriate to attempt to replicate our finding by using rs-fMRI and personality data associated with the (much larger) 3T dataset, which included 4 resting state runs per person. We followed the same procedures as in our initial analysis to obtain a transition rate for each participant in this new dataset. Bootstrap correlations revealed that average transition rate was again modestly, but robustly, correlated with neuroticism, $r(970) = 0.09$, $p = .006$, 95% CI: [0.02, 0.15], thereby confirming our initial findings in a powerful second sample.

In addition, we note that results were much less marginal with respect to the relationship between neuroticism and our other key dependent variable, conformity, still further solidifying the relevance of neuroticism to our transition metrics. As with transition rate, we hypothesized that a noisier mind would result in lower conformity, as the individual would have more idiosyncratic, internally triggered transitions compared to the group. Although the 3T dataset did not contain the necessary mv-fMRI data to replicate this 7T finding, the result was also associated with a higher level of statistical significance, $p = 0.008$, which would survive multiple corrections comparison.

These results and new figure have been added to the text (p. 13).

Supplementary Fig. 5. Higher transition rate correlated to higher trait neuroticism. Transition rate is calculated as the total number of transitions divided by the time in minutes. Neuroticism scores are obtained from the NEO Five Factor Inventory administered to HCP participants.

Reviewer #3, Comment 4

It is also unclear whether other personality traits showed an association with brain measures of transition and stability or whether the results were only specific to neuroticism itself. The authors declare that they have tested the other behavioural traits from the NEO-PI but they only reported the effects of neuroticism. Even if null, the findings for the other personality traits should be included for the sake of clarity and completeness. Furthermore, I wonder whether the statistical model that tested for the associations with personality traits included all the traits simultaneously to account for potential confounding effects driven by the other traits. Or were separate models used for each trait?

Although neuroticism was the only personality trait for which we had explicit hypotheses, we agree that it is appropriate to include the others for clarity and completeness. For descriptive purposes, we now report results describing the relationship between transition rate and each of the personality traits (Supplementary Fig. 6). To maximize statistical power, we focused on the large 3T dataset. With respect to potentially confounding effects, we isolated each personality trait by controlling for all other traits in this descriptive analysis. We found that openness was negatively associated with transition rate (see response to Comment 7 for further comments). These results are now reported in the supplementary text (p. 37).

Supplementary Fig. 6. Full results of correlations between transition rate derived from 3T resting state data and NEO-FFI traits. Correlations were obtained through bootstrapping procedures.

Reviewer #3, Comment 5

Were there gender differences in the main findings or was there an interaction between personality traits and gender? There is a rather robust literature showing that neuroticism levels tend to be higher in females than males. This is mirrored by the higher risk of women to develop anxiety- and depression-related disorders. So I wonder whether the neural correlates of meta-state transitions and stability also depend on gender differences and whether there is any personality by gender interactive effect.

We thank the reviewer for the interesting suggestion. Our results were mixed with respect to this hypothesis. We have incorporated the analysis into our supplemental section as follows (p. 36-37):

Because it is well established that across cultural contexts, neuroticism tends to be higher in females than males (Lynn & Martin, 1997), we organized the 7T dataset by sex, and evaluated the difference in the slope of the correlations established for males versus females (no participants were marked as “other”). We found that neuroticism was associated with higher transition rate in females than in males, $r_{diff} = 0.28$, 95% CI: [0.00 0.57]. However, this pattern did not survive translation to the 3T dataset, $r_{diff} = 0.08$, 95% CI: [-0.05 -.21], providing mixed evidence for the sex specificity of the relationship.

Reviewer #3, Comment 6

The authors do not seem to report the brain localization of the effects of neuroticism on the meta-stable brain measures. Did the regions implicated in transitions show even more transitions in people scoring higher in neuroticism? And the reverse for the regions implicated in meta-stability? Can they run a region of interest approach to address this issue?

It is important to note that our analysis is inherently a “whole-brain” analysis, depending as it does on the combined level of activation of each of 15 networks – there is therefore no way to attribute more or fewer transitions to any one region. However, we believe the closest approximation to this analysis would be to ask: are any of the 15 networks individually responsible for transitions (using the same approach as in the movie feature analysis)?. To conduct this analysis, we treated the timeseries of each network as its own feature vector. Average feature values at transition and meta-stable timepoints were calculated for each participant and each run, then fed into a bootstrap analysis to investigate association with transition versus stability. Our results indicated that no single network was aligned to transitions. Rather, a mixture of networks were associated with transitions, meta-stability, or both/neither. These findings are now integrated into the supplementary text (p. 31).

Another possible approximation of the requested analysis is to explore whether neuroticism predicts the degree to which transition and meta-stable timepoints can be

distinguished at particular ROI's. We tested this idea using both the mv-fMRI and rs-fMRI data, probing locations of peak activation in transition regions (anterior and posterior cingulate, insula), as well as the amygdala (associated with meta-stability). In none of these regions did we find neuroticism to be a predictor; therefore, we did not pursue this line of investigation further.

Reviewer 3, Comment 7

Finally, I suggest the authors to discuss the findings of other papers in the field of personality neuroscience. For example, a recent study (Time-resolved connectome of the five-factor model of personality., Sci Rep. 2019 Oct 21;9(1):15066. doi: 10.1038/s41598-019-51469-2) found that conscientiousness, a trait that is often negatively correlated to neuroticism, is linked to more stable (i.e., reduced time-variant) connectivity patterns across a series of resting-state networks. This brings back the question of whether other traits over and above neuroticism are linked to meta-stable brain measures. Even if this is not the case, the above study could still form the basis of a working hypothesis for the current study which is at the moment not clearly specified.

As suggested, we have elaborated on our results by bringing in information from the descriptive analysis of personality traits we have presented in comment 4. We now add the following discussion to our supplemental results (p. 36):

Among the other traits, we observed only one significant relationship, namely, a positive correlation between openness and transition rate (Supplementary Fig. 6). This link could possibly reflect the relationship that has been demonstrated between this trait and transliminality (Thalbourne, 2000), defined as a "tendency for psychological material to cross thresholds into or out of consciousness" (Thalbourne & Houran, 2000). More frequent crossing of this threshold into consciousness, would, in theory, align with the more frequent arrival of new thoughts. Based on the ideas presented here, this, in turn, should lead to a higher transition rate. However, our interpretation is speculative and post hoc. We therefore suggest the relationship is best regarded as evidence supporting hypothesis generation for future research.

***REVIEWERS' COMMENTS:

Reviewer #1 (Remarks to the Author):

I thank the authors for their extensive response to my comments. They have fully addressed my concerns and I have no additional comments.

Reviewer #2 (Remarks to the Author):

The authors are to be commended for a diligent and thorough revision. All of my comments have been addressed.

Reviewer #3 (Remarks to the Author):

I'm happy with the way the authors have addressed my comments. This is a great paper.

BW

Luca Passamonti